# Structural specificities of cell surface β-glucan polysaccharides determine commensal yeast mediated immuno-modulatory activities

Changhon Lee[1], Ravi Verma[1,12], Seohyun Byun [1], Eun-Ji Jeun[1], Gi-Cheon Kim[2], Suyoung Lee[1], Hye-Ji Kang[3,4], Chan Johng Kim[1], Garima Sharma[1,12], Abhishake Lahiri[5], Sandip Paul [5,13], Kwang Soon Kim[1], Dong Soo Hwang [1,6], Yoichiro Iwakura [7,8], Immacolata Speciale [9,10], Antonio Molinaro [10,11], Cristina De Castro[9,11], Dipayan Rudra [1,12✉] & Sin-Hyeog Im [1,12✉]

Yeast is an integral part of mammalian microbiome, and like commensal bacteria, has the potential of being harnessed to influence immunity in clinical settings. However, functional specificities of yeast-derived immunoregulatory molecules remain elusive. Here we find that while under steady state, β-1,3-glucan-containing polysaccharides potentiate pro-inflammatory properties, a relatively less abundant class of cell surface polysaccharides, dubbed mannan/β-1,6-glucan-containing polysaccharides (MGCP), is capable of exerting potent anti-inflammatory effects to the immune system. MGCP, in contrast to previously identified microbial cell surface polysaccharides, through a Dectin1-Cox2 signaling axis in dendritic cells, facilitates regulatory T (Treg) cell induction from naïve T cells. Furthermore, through a TLR2-dependent mechanism, it restrains Th1 differentiation of effector T cells by suppressing IFN-γ expression. As a result, administration of MGCP display robust suppressive capacity towards experimental inflammatory disease models of colitis and experimental autoimmune encephalomyelitis (EAE) in mice, thereby highlighting its potential therapeutic utility against clinically relevant autoimmune diseases.

[1] Division of Integrative Biosciences and Biotechnology, Department of Life Sciences, Pohang University of Science and Technology (POSTECH), Pohang, Republic of Korea. [2] Department of Microbiology and Immunology, Yonsei University College of Medicine, Seoul, Republic of Korea. [3] Advanced convergence, Handong Global University, Pohang, Republic of Korea. [4] HEM, Pohang, Republic of Korea. [5] Division of Structural Biology and Bioinformatics, CSIR-Indian Institute of Chemical Biology, Kolkata, India. [6] Division of Environmental Science and Engineering, Pohang University of Science and Technology (POSTECH), Pohang, Republic of Korea. [7] Center for Animal Disease Models, Research Institute for Biomedical Sciences, Tokyo University of Science, Noda-shi, Chiba, Japan. [8] Center for Experimental Medicine and Systems Biology, Institute of Medical Science, the University of Tokyo, Minato-ku, Tokyo, Japan. [9] Department of Agricultural Sciences, University of Napoli, Portici, Italy. [10] Task Force on Microbiome Studies, University of Naples Federico II, Naples, Italy. [11] Department of Chemical Sciences, University of Napoli, Napoli, Italy. [12] Present address: ImmunoBiome Inc, Pohang, Republic of Korea. [13] Present address: JIS Institute of Advanced Studies and Research, JIS University, Kolkata, India. ✉email: rudrad@postech.ac.kr; iimsh@postech.ac.kr

The human body hosts an array of microorganisms which constantly interact with its immune system[1] and engage in a symbiotic relationship with the host, aiding diverse processes like digestion, behavior, as well as maturation of immunity[2]. In addition to bacteria, archea, and viruses, fungi also comprise an integral part of the human microbiome where they influence the host's immune system[3]. Innate immune cells detect diverse pathogen-associated molecular patterns (PAMPs) on the surface of fungal cells, including polysaccharides, through pattern recognition receptors (PRRs) such as Toll-like receptors (TLRs) or C-type lectin receptors (CLRs)[4]. Upon detecting signals, innate immune cells change their gene expression profiles and produce cytokines to orchestrate adaptive immunity[5,6].

Yeast-derived polysaccharides have immunomodulatory functions and are broadly utilized as therapeutic agents for activating or suppressing the immune system[7,8]. β-Glucan is the most abundant polysaccharide of the fungal cell wall, which come in complex and diverse structures[9,10]. β-Glucan is known to modulate biological responses that ultimately influence immune responses[11]. Since in general it is known to enhance pro-inflammatory responses, researchers have applied β-glucan as therapy for infectious disease or as an adjuvant for cancer therapy[12,13]. On the other hand, a relatively fewer studies also suggest an anti-inflammatory role of β-glucan in immune responses[14,15]. Similar to fungal β-glucan, fungal mannan is also reported to play immunomodulatory roles[7,16,17]. However, although both pro- and anti-inflammatory roles of yeast-derived polysaccharides have been identified; precise structural specifications and mechanisms by which fungal polysaccharides modify the immune system remain largely unknown.

Zymosan is ghost cell surrogate of the yeast, *Saccharomyces cerevisiae*, which has similar cell wall components like yeast cells[18]. Yeast cell walls and zymosan comprise of two different layers. Outer layer is made up of mannan in conjugation with cell wall proteins and inner layer is composed of β-glucan as dominant component that have complex structure primarily consisting of multiple layers of β-1,3-linked glucan connected through relatively less abundant β-1,6-glucan-containing polysaccharides and chitin[19,20]. Many studies utilized zymosan to investigate the role of fungal polysaccharides on the immune system. Zymosan is a potent immune stimulator that, upon binding to PRRs of innate immune cells like dendritic cells and macrophages, promotes the release of pro-inflammatory cytokines such as IL-6, TNF-α, and IFN-γ. In light of these findings zymosan have been demonstrated to largely exacerbate inflammatory disorders[20–23]. In addition to these studies, some other reports, however, revealed controversial results that demonstrate that zymosan can induce immunological tolerance by inducing IL-10-producing tolerogenic antigen-presenting cells that suppress antigen-specific T-cell responses to ameliorate autoimmune disease[24,25].

Here, by using zymosan as well as purified yeast extract, we systematically dissect yeast cell wall-derived polysaccharide components and investigate their roles under in vitro and in vivo immune-modulatory conditions. We find that under steady-state conditions the abundance of yeast cell wall-derived polysaccharides composed of β-1,3-glucan potentiates pro-inflammatory effects. However, upon its removal by enzymatic interventions or through purification techniques, a relatively minor class of cell surface polysaccharides is unmasked that is capable of exerting potent anti-inflammatory effects on the immune system. We named this mannan/β-1,6-glucan-containing polysaccharides (MGCP), which by facilitating the induction of Treg cells as well as by directly suppressing the IFN-γ producing Th1 cell differentiation, are capable of exerting potent immunosuppression toward multiple models of experimental autoimmune diseases. Importantly, mechanistic and structural specificities of MGCP highlight its therapeutic applicability against autoimmune diseases.

## Results

**The major pro-inflammatory component of zymosan is β-1,3-glucan.** To examine immunomodulatory functions of yeast polysaccharides, we treated DCs with zymosan, followed by culture with naive CD4+ T cells either in control Th0 condition, or in the presence of suboptimal levels of respective T helper cell skewing cytokines. Zymosan primed DCs promoted the induction of IFN-γ under Th1 skewing conditions. Interestingly this effect, albeit moderate, was specifically observed under stringent Th1-promoting conditions. Control experiments conducted under Th0 conditions on the other hand, displayed substantially reduced baseline expression level of IFN-γ that remained unaltered upon culturing with zymosan primed DCs (Figs. 1a and S1a). Unlike Th1, induced Treg (iTreg) and Th17 generation, as determined by Foxp3 and IL-17 expression, respectively, remained largely unaffected (Figs. 1a, S1a–c).

In order to determine the precise roles of the major component of zymosan on immune responses, we removed β-1,3-glucan from zymosan, using a specific cleaving enzyme. Interestingly, elimination of β-1,3-glucan from zymosan upon treatment with β-1,3-glucanase led to a significant increase in the induction of Treg cells (Fig. 1b). On the other hand, the Th1-promoting property of zymosan was severely compromised (Fig. 1c). In order to ascertain the functional properties of zymosan-derived β-1,3-glucan in the context of anti-tumor immunity, we treated B16. F10 tumor-bearing mice with either zymosan or β-1,3-glucan depleted zymosan every alternate day throughout the course of the experiment. Consistent with a previous report[26], tumor growth and size were significantly reduced upon treatment with zymosan. However, removal of β-1,3-glucan abolished its anti-tumor properties (Fig. 1d–f). This was accompanied with a reversal in the zymosan-mediated increase in frequencies of IFN-γ+ Th1 cells, as well as a small but significant increase in the frequency of CD4+Foxp3+ Treg cells primarily in the tumor-draining lymph nodes (Fig. 1g, h). Taken together these findings strongly indicated that the pro-inflammatory properties of zymosan that are primarily responsible for promoting Th1 differentiation of effector T cells, are mediated by β-1,3-glucan. These findings further implicated that there may be other polysaccharide components present in yeast cell wall that can promote anti-inflammatory responses by facilitating Treg induction upon enzymatic removal of β-1,3-glucan.

**Cell surface mannan/β-glucan-containing polysaccharides of yeast has immune-regulatory properties.** Results so far indicated that under steady-state conditions, presumably the abundance of β-1,3-glucan in zymosan promotes pro-inflammatory immune response, whereas its inherent immune-regulatory properties are exposed only upon β-1,3-glucanase mediated elimination of β-1,3-glucan. Since the pro-inflammatory properties of zymosan are extensively studied and well documented[21–23], we sorted to focus on the relatively less explored immunoregulatory aspects of yeast-derived cell surface polysaccharides. To this end, in order to be more physiologically relevant, we purified cell wall-derived polysaccharides from yeast extract derived from *Saccharomyces cerevisiae*. Of note, the purification process we employed, selectively purified water-soluble polysaccharides, while β-1,3-glucan, being water insoluble[27], was excluded from the final purified yeast extract (YE) derived polysaccharides (schematically explained in Fig. S2a and in "Methods"). As expected, purified YE significantly facilitated the generation of iTreg cells under suboptimal Treg-inducing conditions. β-1,3-Glucanase treatment only minimally

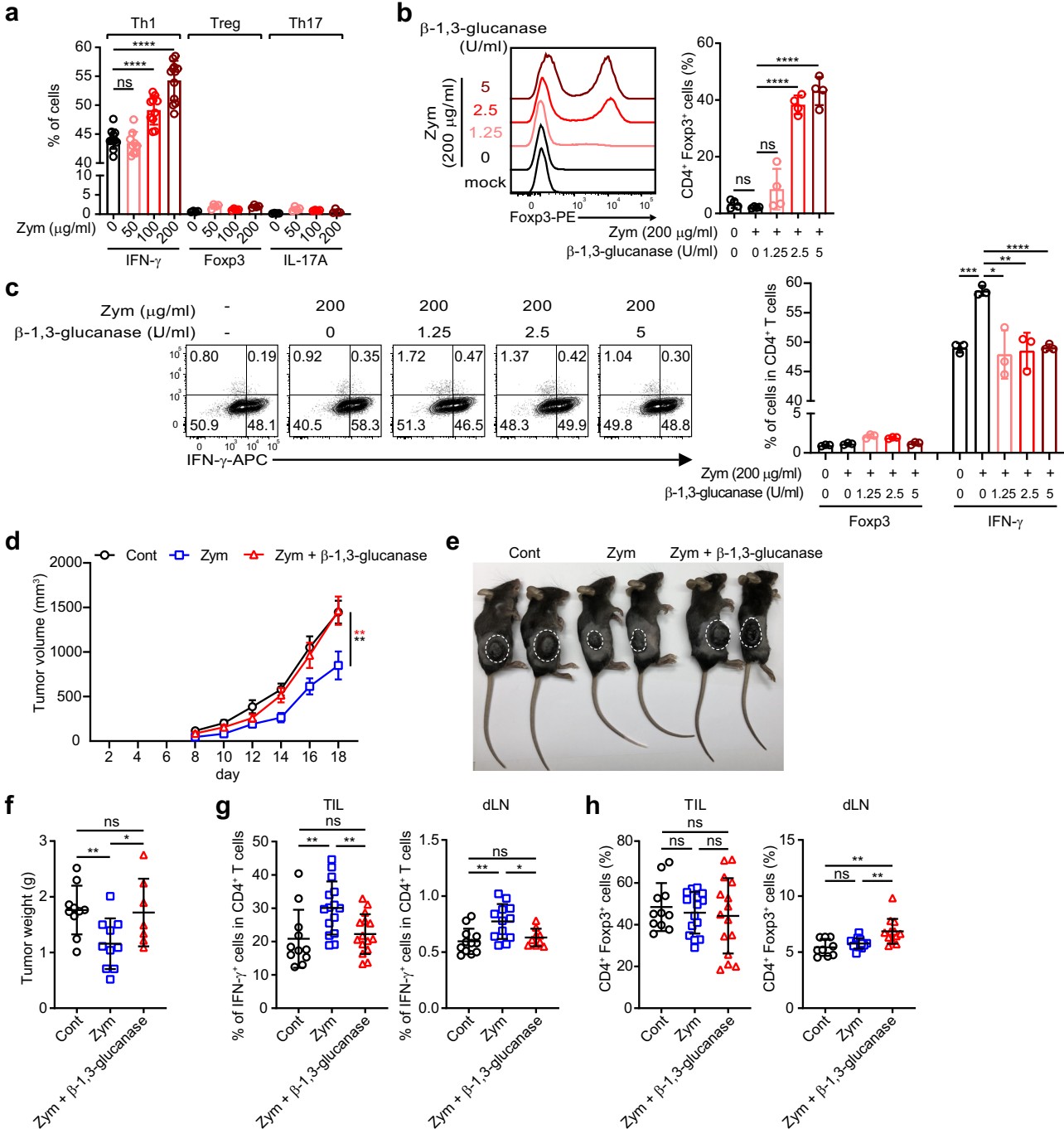

**Fig. 1 β-1,3-Glucan of zymosan enhances Th1 immune response and restrains Treg cell differentiation. a–c** Splenic CD11c[+] DCs were stimulated with zymosan in the presence or absence of zymolase, which has β-1,3-glucanase activity, and cultured with naive CD4[+] T cells under indicated T helper cell skewing conditions with suboptimal amounts of respective cytokines. **a** Frequencies of IFN-γ, Foxp3, and IL-17A producing cells under respective CD4 T-cell skewing conditions. Results are analyzed from 4 independent experiments with minimum 9 samples (IFN-γ) and 2 independent experiments with 4 samples (Foxp3 and IL-17A). Representative flow cytometric plots (left panels) and frequencies (right panels) of Treg (**b**) and Th1 cells (**c**) induced under Treg or Th1 inducing conditions are shown. Each dot represents an individual sample. Results are analyzed with pooled data of 2 samples over 2 independent experiments (**b**) or 2 biologically independent experiments for which representative data is shown (**c**). The graphs show mean ± SD. *$p < 0.05$, **$p < 0.01$, ***$p < 0.001$, ****$p < 0.0001$ (two-tailed student's t test). **d–h** Mice were grafted with 2 × 10[5] cells of B16.F10 melanoma subcutaneously, followed by injection of zymosan or zymosan devoid of β-1,3-glucan every alternate day. Tumor growth (**d**, **e**) and tumor weight (**f**) were measured. Frequency of CD4[+]IFN-γ[+] T cells was analyzed (**g**) among tumor infiltrated lymphocytes (TILs) and tumor-draining lymph nodes (dLNs). CD4[+]Foxp3[+] T-cell frequency (**h**) was analyzed in TILs and dLNs. Tumor growth result is analyzed with minimum 9 individual mice from 2 independent experiments. Tumor weight was measured with minimum 7 mice from 2 independent experiments. Result of CD4[+]IFN-γ[+] T and CD4[+]Foxp3[+] T cell in TIL and dLNs were analyzed with minimum 9 mice from 2 independent experiments. Each dot represents an individual mouse. Graphs show mean ± SEM. **d** **$p < 0.01$ or SD, **f–h** *$p < 0.05$, **$p < 0.01$ (two-tailed student's t test). ns: not significant. Source data are provided as a Source data file.

enhanced this potential, suggesting efficient elimination of β-1,3-glucan from the purified YE (Fig. 2a).

Nuclear magnetic resonance (NMR) analyses (Fig. S2b and S2c, chemical shifts in Table S1) as well as polysaccharide composition (Fig. S2d) defined the nature of the polysaccharides derived from the yeast cell wall, a mannan, and a branched β-1,6-glucan. The relative abundance of the two polysaccharides was 80.6% to 14.9% in favor of mannan, as evaluated by comparing the mono-saccharide composition of the mixture (Fig. S2b–d). Importantly, β-1,3-glucan was undetectable by NMR (Fig. S2b, c), presumably due to its elimination in the water-insoluble fraction. We named these soluble polysaccharides extracted from the yeast cell wall as "mannan/β-1,6-glucan containing polysaccharides" (MGCP in brief). The structure of the mannan component in MGCP was comparable with typical mannan structure found on *S.cerevisiae* cell wall, and presented a α-1,6-linked mannose backbone with part of the units not further substituted (24.8%) or with side chains composed of mono- (30.0%), di- (16.8%) and tri-saccharides (28.4%) connected through α-1,2-linkage to the main backbone (Fig. 2b), as inferred by NMR analysis and integration of the appropriate signals. β-Glucan fraction in MGCP consisted in a β-1,6-linked glucose backbone, with 18% of the units branched with a single glucose linked through β-1,3-linkage (Fig. 2c). To determine its immunological properties, we next assessed the in vitro Treg and Th1 inducing capabilities of MGCP. Strikingly, MGCP primed splenic DCs not only promoted iTreg cell differentiation (Fig. 2d), it also considerably repressed Th1 cell generation in a dose-dependent manner (Fig. 2e).

In order to determine the essential component of MGCP responsible for iTreg cell differentiation, we fractionated MGCP according to molecular weight through liquid chromatography with gel filtration column. Chromatogram profile of MGCP showed two distinguishable peaks with high (>50 kDa) and low (<50 kDa) molecular weight (M.W.) (Fig. S3a). Further analysis by gas chromatography and mass spectrometry (GC-MS) revealed that the lower M.W. fraction comprises a mixture of both mannan and β-1,6-glucan. On the other hand, the high M.W. fraction primarily consisted of mannan, where β-1,6-glucan was not present or below the instrumental detection limits (Fig. S3a, b). Interestingly, when both fractions were assayed, only the low M.W. fraction which retains β-1,6-glucan significantly induced Treg cells to an extent comparable to purified MGCP (Fig. 2f). In agreement to this finding, enzymatic elimination of β-1,6-glucan from MGCP with the cognate cleaving enzyme β-1,6-glucanase led to significant reduction in its Treg-inducing as well as Th1 repressing potential (Fig. 2g, h). Taken together these results strongly indicated that the immune-regulatory properties of yeast cell surface polysaccharides can be primarily attributed to the activity of β-1,6-glucan of MGCP.

In a study published recently, we identified components of CSGG, isolated from the commensal microbiome *B. bifidum* to have similar Treg-inducing properties, and the key effector component of CSGG responsible for this activity was also identified to have a backbone comprising of β-1,6-glucan polymers[28]. However, β-1,6-glucan derived from MGCP, with a side chain of a single β-1,3-linked glucose, appeared to differ from that derived from CSGG, where the side chain glucose molecule is linked with the backbone through α-1,2-linkage (Fig. S3c). These structural specificities suggested that distinct functional differences may exist between MGCP and CSGG. Indeed, when tested side by side, only MGCP and not CSGG were able to repress Th1 cell generation under Th1 skewing condition in vitro (Fig. S3d). These results suggested that while both MGCP and CSGG can facilitate iTreg induction, MGCP is capable of exerting its immune-regulatory effect by suppressing differentiation of

pro-inflammatory effector T cells as well, a property that is absent in CSGG.

'Depleted Zymosan' is a well-studied variety of zymosan, in which hot alkali treatment results in the removal of Toll-like receptor 2 (TLR2) stimulating pro-inflammatory property of zymosan[22]. Provided that MGCP share similarly anti-inflammatory properties with depleted zymosan, we decided to compare the polysaccharide composition of MGCP with that of two different zymosans. We observed that mannan is present in MGCP and zymosan, while absent (or below the detection limit) in depleted zymosan (Fig. S4a). Linkage analysis to differentiate the β-glucan components in zymosan revealed that β-1,3-glucan was over 98% of the mixture, while β-1,6-glucan was only ~1.3% comprising of β-1,6-glucan (Fig. S4b–d, values calculated by the comparison of the 3-Glc and 6-Glc ratios). Unlike zymosan, the proportion of β-1,6-glucan is increased in depleted zymosan to 20% while β-1,3-glucan is decreased to nearly 80% (Fig. S4b–d). Finally, in agreement with the NMR analysis, the glucan component detected by linkage analysis of MGCP was β-1,6-glucan only, namely MGCP was completely devoid of β-1,3-glucan (Fig. S4b–d).

We next determined the functionality of MGCP-induced iTreg cells in vivo. Sorted congenically marked CD45.1+ iTreg cells generated in the presence of MGCP primed DCs were tested for their suppressive capacity in a well-established cell transfer model of colitis[29] upon co-transfer in lymphopenic host *Rag1−/−* mice along with CD45.2+CD4+Foxp3−CD62Lhi naive T cells. While transfer of naive T cells alone, in the absence of Treg cells, induced characteristic body weight loss, shortening of colon as well as increased histopathologic score, all of which are indicative of colitogenic pathophysiology, co-transfer of MGCP-induced iTreg cells efficiently prevented the disease (Fig. 3a–c). Furthermore, mice harboring MGCP-induced iTreg cells also displayed considerable reduction in IFN-γ expressing Th1 cells from the co-transferred CD4+Foxp3− effector T-cell compartment in large intestine lamina propria compared to the 'naive T-cell transfer only' group (Fig. 3d). Taken together these results suggested that MGCP-induced iTreg cells were functionally active to suppress the differentiation of inflammatory Th1 cells and can ameliorate inflammatory colitis.

## MGCP treatment ameliorates experimental autoimmune diseases by promoting de novo Treg cell induction and suppressing Th1 cell generation in vivo.

We next examined whether direct administration of MGCP can promote anti-inflammatory responses in vivo. Mice were orally administered with MGCP followed by transfer of CD4+CD45.1+ naive T cells (Fig. 4a). Under such conditions, as expected, only a small proportion of the transferred CD45.1+ T cells could be recovered from the colon and small intestine of recipient mice in the end of the experiment presumably due to competition from the already occupied T-cell niche in lymphoreplete recipients (Fig. 4b, c left panels). Nevertheless, among the small number of cells recovered, recipients pre-treated with MGCP displayed significantly enhanced capacity of iTreg induction compared to mock-treated mice (Fig. 4b, c right panels). A similar trend, while not statistically significant within the cohort examined, was also observed among the recipient CD4+CD45.1− T cells in the MGCP-treated group that displayed an increase in Treg frequency and number (Fig. S5a, b). Furthermore, frequency of Foxp3+Helios− Treg cells was increased in the MGCP-treated recipient compartment, presumably as a consequence of increased iTreg generation (Fig. S5c, d). In addition to facilitating iTreg induction, MGCP administration was also found to significantly reduce differentiation of IFN-γ expressing effector CD4+Foxp3− T cells from

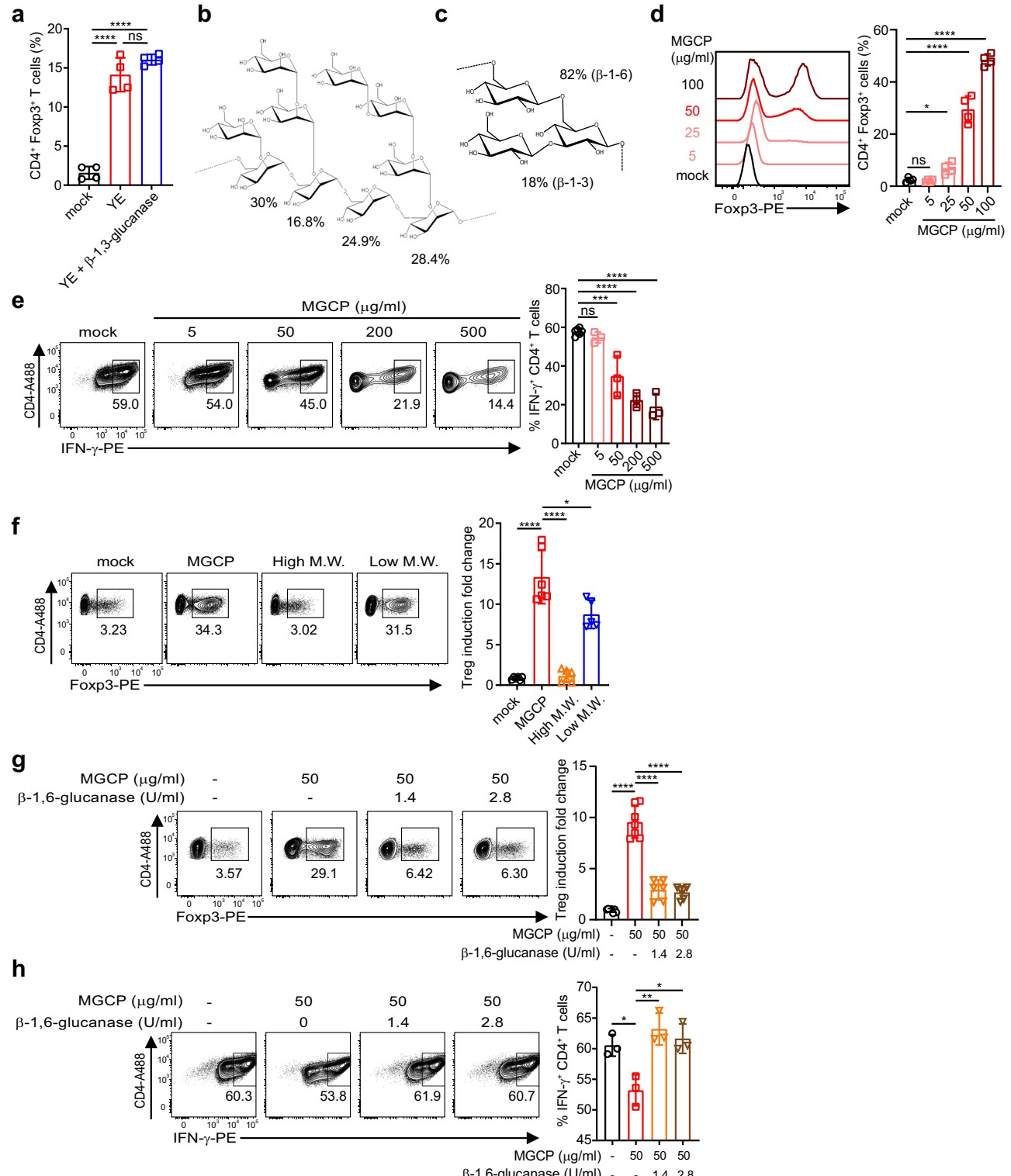

**Fig. 2 Yeast cell wall-derived MGCP facilitates Treg cell induction and represses Th1 differentiation. a** Yeast extract (YE) was purified from intact or β-1,3-glucan removed yeast cells. Splenic CD11c+ DCs treated with these YE were cultured with naive CD4+ T cells in the presence of minimal amount of Treg-inducing cytokines. Frequency of Treg cells is shown. Each dot represents an individual sample. Data are analyzed with pooled data of 2 independent experiments. **b, c** Structural analyses of MGCP through nuclear magnetic resonance (NMR). Chemical structure of mannan (**b**) and β-1,6-glucan moiety of MGCP (**c**). **d–h** Splenic CD11c+ DCs were subjected to MGCP (**d, e**), individual MGCP fractions (**f**) or enzymatic treatment as indicated (**g, h**), followed by culture with naive CD4+ T cells. Representative flow cytometric plots (left) and frequency (right) of Treg cells in the presence of suboptimal Treg-inducing condition (**d, f, g**) and Th1 cells under suboptimal Th1 skewing condition (**e, h**) are shown. Each dot represents an individual sample. Results are analyzed from pooled data of 3 independent experiments (**d, f**) or 2 independent experiments (**g**) and 2 independent experiments presented with representative data (**e, h**). All graph plots show the mean ± SD. *$p < 0.05$, **$p < 0.01$, ***$p < 0.001$, ****$p < 0.0001$ (two-tailed student's $t$ test). ns: not significant. Source data are provided as a Source data file.

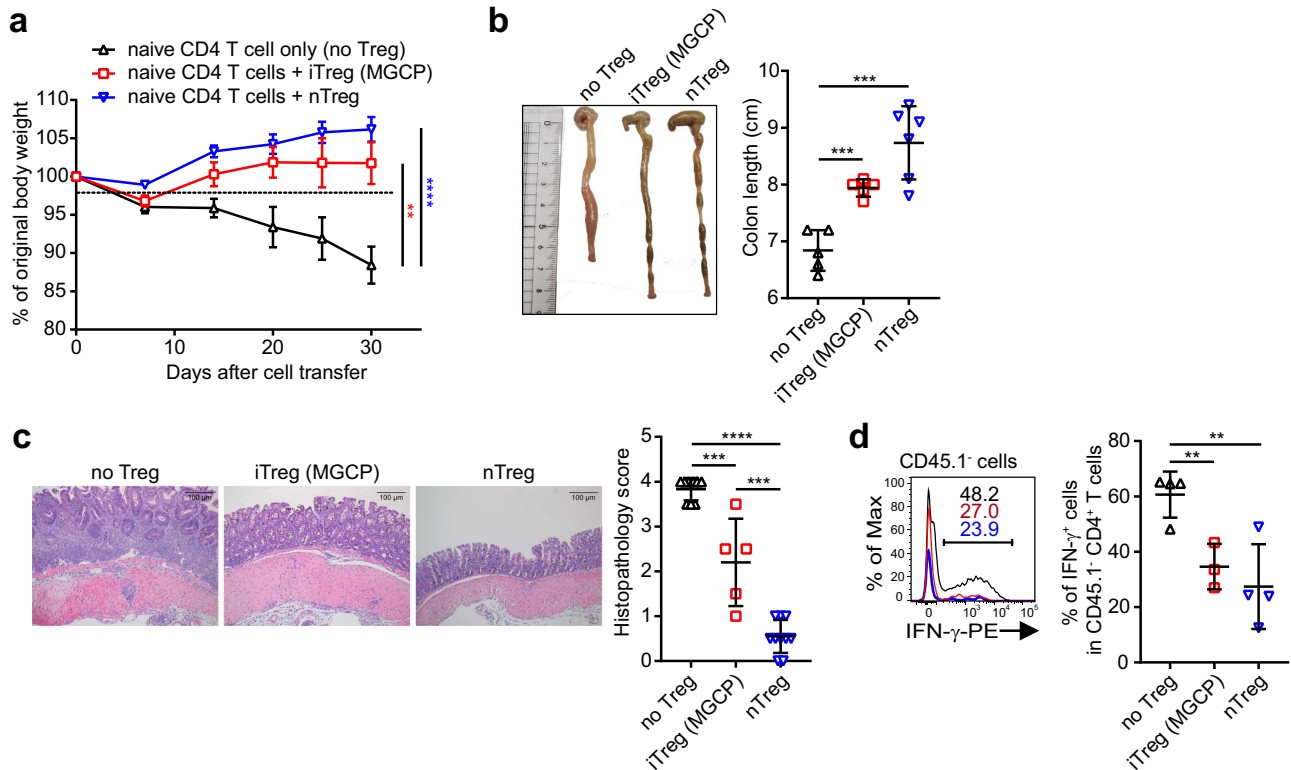

**Fig. 3 MGCP promotes functionally active Treg cell differentiation. a–d** iTreg cells were generated in vitro by co-culture of MGCP-treated splenic DCs with naive CD45.1+CD4+ T cells (CD45.1+CD4+Foxp3−CD44loCD62Lhi). iTreg cells (CD45.1+CD4+Foxp3+) were FACS sorted and adoptively transferred into $Rag1^{−/−}$ mice along with naive CD4+ T cells (CD45.2+CD4+Foxp3Thy1.1CD44loCD62Lhi). CD4+ naive T cells alone were transferred as controls (no Treg group). Colitis development and severity were assessed by body weight change (**a**) and changes in colon length (**b**). Representative colon sections were stained with H&E (**c**) and histopathology score was measured. Frequency of IFN-γ+ donor naive CD4+ T cells (CD45.2+) was assessed (**d**) in colonic lamina propria. Each dot represents an individual mouse. Data are analyzed from 4 independent experiments with minimum 10 mice (**a**). Results are analyzed with pooled data from 2 independent experiments with minimum 5 mice (**b, c**). Graph is presented with representative data of 2 independent experiments with minimum 3 mice (**d**). Graph plots show the mean ± SEM. **a** **$p < 0.01$, ****$p < 0.0001$ or ±SD. **b–d** **$p < 0.01$, ***$p < 0.001$, ****$p < 0.0001$ (two-tailed student's t test). Source data are provided as a Source data file.

intestinal lamina propria (Fig. 4d, e). Taken together these findings strongly establish that MGCP exerts its immunoregulatory effects on one hand by enhancing anti-inflammatory parameters like facilitating iTreg generation and on the other hand by efficiently suppressing differentiation of deleterious IFN-γ expressing CD4+ effector T cells.

Next, to assess whether MGCP induce antigen-reactive Treg cells and suppress inflammatory immune responses in vivo, we utilized CBir mice, which possess microbial flagellin-reactive CD4+ T cells[30]. Naive CD4+ T cells from CBir mice were adoptively transferred into $Rag1^{−/−}$ mice, followed by administration of either mock (DW) or MGCP every other day through oral route (Fig. 5a). As expected, adoptive transfer of microbe responsive naive CD4+ T cells resulted in the development of experimental colitis through excessive immune responses against commensal microbiota[31]. However, MGCP supplementation rendered significant resistance against body weight loss (Fig. 5b). Furthermore, MGCP administration prevented shortening of colon length, rendered protection against hyperplasia of epithelial cells (Fig. 5c, d) and alleviated overall pathological score compared to the mock-treated mice (Fig. 5e). This was accompanied with increase in colonic Treg cells and concomitant reduction in IFN-γ producing CD4+Foxp3− population (Fig. 5f, g). Percentage of IL-17 producing Th17 population remained unchanged (Fig. S6a). On a per cell basis, the expression of CD103 and CTLA-4, two molecules implicated in suppressive activity of Treg cells, remained unchanged (Fig. S6b). These data demonstrated that

administration of MGCP alleviated the development of the inflammatory colitis primarily by promoting the generation of Treg cells in response to commensal antigen as well as by suppressing induction of pathogenic Th1 immunity.

In order to determine whether MGCP is protective against a second model of experimental autoimmune disease, we applied MGCP to mice immunized with MOG35–55 peptide to induce experimental autoimmune encephalomyelitis (EAE). MOG35–55 immunized mice were injected with distilled water (mock) or MGCP daily till the end of the experiment (Fig. 5h). MGCP treatment almost completely prevented mice from developing EAE compared to mock-treated group and suppressed immune cell infiltration into the spinal cord (Fig. 5i–k). In accordance, frequency of IFN-γ+IL-17A+ double-positive CD4+ T cells implicated in EAE pathology were significantly suppressed in spinal cord of MGCP-treated mice compared mock-treated mice (Fig. 5l). IFN-γ and IL-17A producing single-positive CD4+ T cells were also reduced in both spinal cord and draining lymph node of MGCP administered mice compared to mock (Fig. S6c–e). Treg cell frequency, while comparable in spinal cord, was significantly increased in draining lymph nodes of MGCP-treated mice compared to mock (Fig. S6f, g). Furthermore, MGCP treatment reduced frequency of spinal cord infiltrating monocyte, but macrophage and monocyte-derived DCs (MoDCs) were unaffected (Fig. S6h).

A recent study reported that one of the mechanisms which antigen-specific Treg cells employ for suppressing effector cell

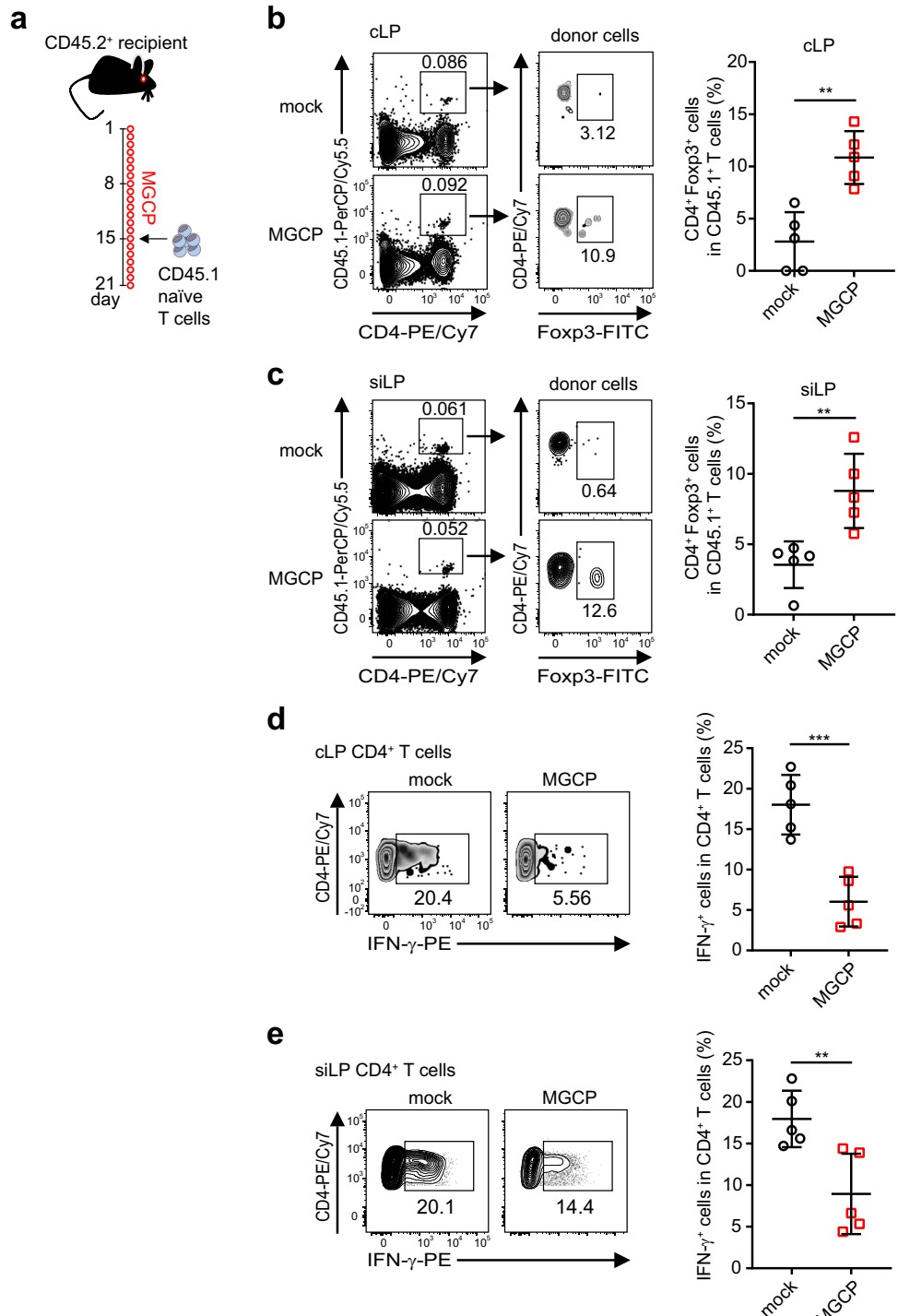

**Fig. 4 MGCP administration in vivo promotes de novo iTreg generation and suppresses IFN-γ production. a** Experimental scheme. Mice were administered either mock (DW) or MGCP every day for 2 weeks prior to receiving congenically marked naive CD4+ T cells (CD45.1+CD4+Foxp3−CD44loCD62Lhi). MGCP was administered for 1 additional week after cell transfer, after which intestinal CD4+ T cells were analyzed. **b, c** Representative flow cytometric plots (left panel) and frequencies (right panel) of de novo generated Treg cells originated from donor cells (CD45.1+CD4+) in lamina propria of colon (**b**) and small intestine (**c**). **d, e** Representative flow cytometric plot (left panel) and frequency (right panel) of IFN-γ+ effector CD4+ T cells (CD4+Foxp3−) in colon (**d**) and small intestine (**e**). Each dot represents individual mouse. All data are analyzed with pooled data from 2 independent experiments with 5 mice. All graphs show the mean ± SD. **p < 0.01, ***p < 0.001 (two-tailed student's t test). Source data are provided as a Source data file.

differentiation, is to directly bind and down-modulate the surface expression of MHCII on DCs[32]. Provided that MGCP treatment in our EAE modeled to such strong repression of disease induction, and near-complete elimination of both Th1 and Th17, as well as IFN-γ+IL-17+ double-positive cells, it appeared

possible that due to Treg-mediated down-modulation of MHCII from DC surface, the antigen-specific T cells were unable to get access to cognate peptide-MHCII complex, hence are unable to differentiate. On the contrary, when DCs in the draining lymph nodes of EAE induced mice were analyzed, MHCII expression

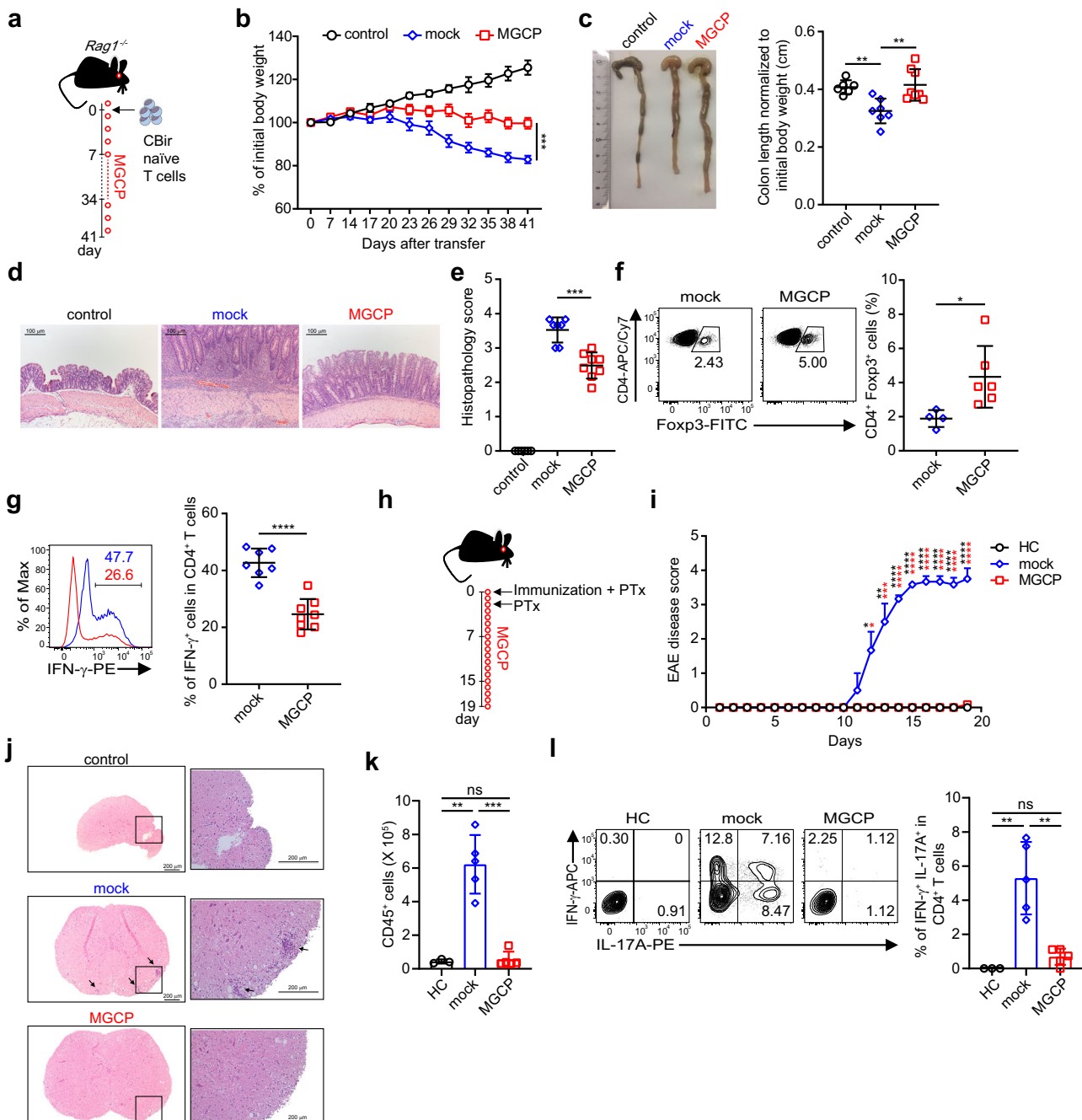

**Fig. 5 MGCP supplementation ameliorates experimental inflammatory diseases. a** Schematic of experimental model of colitis employed. Naive CD4+ T cells (CD4+Foxp3−CD44loCD62Lhi) isolated from CBir mice were adoptively transferred into *Rag1* deficient mice. MGCP, or distilled water as mock, was fed every other day from a day before transferring the cells and continued for entire experimental period. **b** Body weight change as a measure of colitis development is shown. Data are analyzed with minimum 6 mice from 2 independent experiments. **c** Representative image of colon (left) and colon length normalized to initial body weight (right). Each dot represents individual mouse. **d** Representative colon histology image after hematoxylin and eosin (H&E) staining. **e** Histopathology score of colitis. **f** Representative FACS plots (left) and frequencies (right) of CD4+Foxp3+ T cells in colon. **g** Representative histogram (left) and frequency (right) of IFN-γ producing colonic effector CD4+ T cells (CD4+Foxp3−). Data are analyzed from pooled data with minimum 6 mice (**c**, **e**) and 7 mice (**g**) or representative result with minimum 4 mice (**f**) of 2 independent experiments. Each dot represents an individual mouse. **h** Scheme of EAE experiment. Mice were immunized with MOG35-55 and complete Freund's adjuvant (CFA) to develop EAE as described in "Methods". Immunized mice were injected daily with MGCP, or distilled water as mock, from the day of immunization. **i** Graphical display of EAE clinical scores for each treatment groups. Data are presented with representative result of 2 independent experiments with minimum 3 mice. Statistical significance was measured by one-way ANOVA with Turkey's multiple comparison test. **j** Representative spinal cord sections stained with H&E. Magnifications, ×5 (left) and ×20 (right). Arrows indicate infiltrated lymphocytes. **k** Absolute numbers of infiltrating lymphocytes in spinal cords of the different treated groups. Data are presented with representative result of 2 independent experiments with minimum 3 mice. **l** Representative flow cytometric plots (right) and percentage (left) of CD4+IFN-γ+IL-17A+ T cells in the spinal cords of indicated groups of mice. Each dot represents an individual mouse. Data from one of three independent experiments with minimum 3 mice is shown. Graphs show mean ± SEM. **b**, **i** *$p < 0.05$, **$p < 0.01$, ***$p < 0.001$, ****$p < 0.0001$ or ±SD. **c**, **e**, **f**, **g**, **k**–**l** *$p < 0.05$, **$p < 0.01$, ***$p < 0.001$, ****$p < 0.0001$ (two-tailed student's *t* test). ns: not significant. Source data are provided as a Source data file.

was found to be moderately increased in the MGCP-treated group (Fig. S6i). Taken together, these results suggested that administration of MGCP restrained inflammatory autoimmune diseases by facilitating Treg cell induction as well as by suppressing the generation of pathologic cytokine-producing immune cell populations.

**Immune suppression by MGCP is a DC-dependent Cox2-mediated process.** We next sorted to determine the mechanism(s) by which MGCP facilitate iTreg induction from naive T cells under suboptimal Treg-inducing condition. We first assessed whether the Treg-inducing property of MGCP is a DC-dependent or a DC-independent mechanism. Since a previous report[33] has demonstrated that monomeric D-(+)-mannose is capable of facilitating Treg differentiation in a DC-independent manner by directly acting on naive CD4$^+$ T cells, we used D-(+)-mannose as a control in this assay. Our results clearly suggested that the beneficial effects of MGCP on Treg induction are only manifested when DCs pretreated with MGCP are included in the assay, suggesting that MGCP-mediated phenotypic changes in DCs are an essential prerequisite (Fig. 6a, b). Furthermore, splenic CD8−CD11b$^+$ DCs and their equivalent intestinal CD103$^+$CD11b$^+$ DCs were found to be the key DC subset responsible (Fig. 6c–f). Macrophages displayed significantly enhanced basal Treg-inducing activity, that was marginally enhanced upon pretreatment with MGCP (Fig. S7a). Contrary to MGCP, D-(+)-mannose, as expected was found to facilitate Treg induction by a DC-independent mechanism (Fig. 6b). These results demonstrated that MGCP-mediated Treg induction was mechanistically distinct from that of D-(+)-mannose. Furthermore, it also excluded a formal possibility that any broken down monosaccharide form of D-(+)-mannose from the mannan present in MGCP may be contributing to MGCP-mediated induction of Treg cells.

In order to gain molecular insights on the mechanisms by which MGCP influence DCs to facilitate Treg induction, we treated splenic CD8−CD11b$^+$ DCs with MGCP or mock for various time points and performed high-throughput next-generation cDNA sequencing [RNA sequencing (RNA-seq)]. MGCP treatment enhanced expression of tolerogenic DC-associated markers, such as *Il10*, *Cd274* (encoding PD-L1), *Arginase* (*Arg*), and *Il27*[34,35] (Fig. 6g, h). Intriguingly MGCP significantly enhanced expression of *Ptgs2* (encoding cyclooxygenase-2, Cox2) in DCs compared to mock DCs (Fig. 6h). Also, to confirm whether administration of MGCP modulates transcriptome of intestinal DCs toward tolerogenic phenotype and mimics splenic DCs in vivo, we fed MGCP to germ-free (GF) mice, presuming that the effect of MGCP would be most pronounced under GF conditions devoid of any contribution from commensal yeast species in the mock-treated samples. CD11c$^+$ DCs were isolated from colon of mock or MGCP supplemented mice and RNA-seq was performed. In accordance with splenic CD8−CD11b$^+$ DCs, MGCP treatment enhanced expression of tolerogenic DC-associated markers and increased *Ptgs2* expression in colonic DCs (Fig. S7b, c).

Since Cox2 has been previously described to induce Treg cells in tumor environments[36] but suppress IFN-γ expression and Th1 immunity[37,38], we assessed the role of Cox2 on immune modulation by MGCP. Splenic DCs were stimulated with MGCP, and then cultured with naive CD4$^+$ T cells, in the presence of Celecoxib, a selective inhibitor of Cox2. Induction of Treg cells by MGCP was compromised significantly by inhibiting Cox2 (Fig. 6i, j) under suboptimal Treg skewing environment. On the other hand, Cox2 inhibition recovered differentiation of Th1 cells which was repressed by MGCP in the presence of Th1 skewing condition (Fig. 6k, l). These findings suggested that Treg cell induction and

suppression of Th1 cell differentiation by MGCP were specifically mediated by CD8−CD11b$^+$ DC subset in a Cox2-dependent manner. Furthermore, MGCP alters the DC transcriptome landscape toward a tolerogenic DC phenotype.

Since commensal microbiota is well known for its capability of regulating host immune responses, we also determined whether administration of MGCP could influence intestinal microbiome composition under conditions of colitogenic inflammation. For this, unlike previously employed CBir TCR transgenic naive CD4$^+$ T cells that is dependent on flagellin expressing microbial species for inducing colitis (Fig. 5a–g), we used a polyclonal T-cell transfer model of colitis in which naive CD4$^+$ T cells from wild-type (WT) mice are transferred in *Rag1*$^{-/-}$ lymphodeplete host animals for colitis induction[29]. Principal coordinate analysis (PCoA) analysis generated by beta-diversity using Bray–Curtis dissimilarities revealed that microbiome structure of MGCP-treated mice is similar to that of mice treated with distilled water (mock) during the induction of colitis (Fig. S8a). Also, while as expected there were significant differences in microbial diversity between groups before and after disease induction, the relative abundance of major bacterial taxa in mock and MGCP-treated groups were comparable at the phylum and genus levels in mice analyzed on day 32 after induction of colitis (Fig. S8b, c).

**Dectin1- and TLR2-mediated signaling in DCs differentially facilitate MGCP-dependent iTreg induction and Th1 suppression, respectively.** DCs recognize microbial antigens including polysaccharides present in intestine through diverse PRRs[39,40]. To identify innate immune receptors associated with MGCP-induced immune modulation, we investigated the roles of diverse PRRs that are known to recognize fungal polysaccharides including β-glucan. In particular, we found Dectin1-deficient splenic DCs to be significantly compromised for MGCP-mediated Treg cell induction. Dectin2 deficiency on the other hand was found to have little effect. Furthermore, while in concert to our previous findings[28], TLR2-deficieny in DCs only affected Treg-inducing capacity of CSGG, it did not affect MGCP-mediated Treg induction (Fig. 7a). We also examined the role of other C-type lectin receptors by utilizing respective antagonistic blocking antibodies of DC-SIGN, Mincle, and mannose receptor. Blocking of DC-SIGN signaling in DCs moderately reduced differentiation of Treg cells by MGCP at higher concentration, but blocking other receptors did not show any significant impact (Figs. 7b and S9a).

Based on these findings, we assessed the role of Dectin1 in Treg cell induction by MGCP in vivo. For this, CD4$^+$ T cells from congenically marked OTII mice were transferred in Dectin1 sufficient and deficient mice that were prior supplemented with MGCP for 2 weeks. The recipient mice were fed with Ovalbumin and MGCP for 1 more week, after which iTreg induction was assessed in small intestine (Fig. 7c). Indeed, whereas Ovalbumin fed Dectin1 sufficient mice significantly enhanced iTreg induction compared to mock; this beneficial effect was completely abolished in Dectin1-deficient recipient mice (Fig. 7d). In light of these results, we further tested the role of Dectin1 for MGCP-mediated induction of immune tolerance under disease condition. Dectin1 sufficient or deficient mice were immunized with MOG$_{35-55}$ peptide and injected with distilled water (mock) or MGCP. As expected, MGCP treatment protected the Dectin1 sufficient mice from developing EAE, but this protective property of MGCP was completely abolished in Dectin1-deficient mice (Fig. 7e). In accordance with disease score, MGCP treatment increased the frequency of Treg cells in draining lymph nodes of Dectin1 sufficient, but not in Dectin1-deficient mice (Fig. 7f).

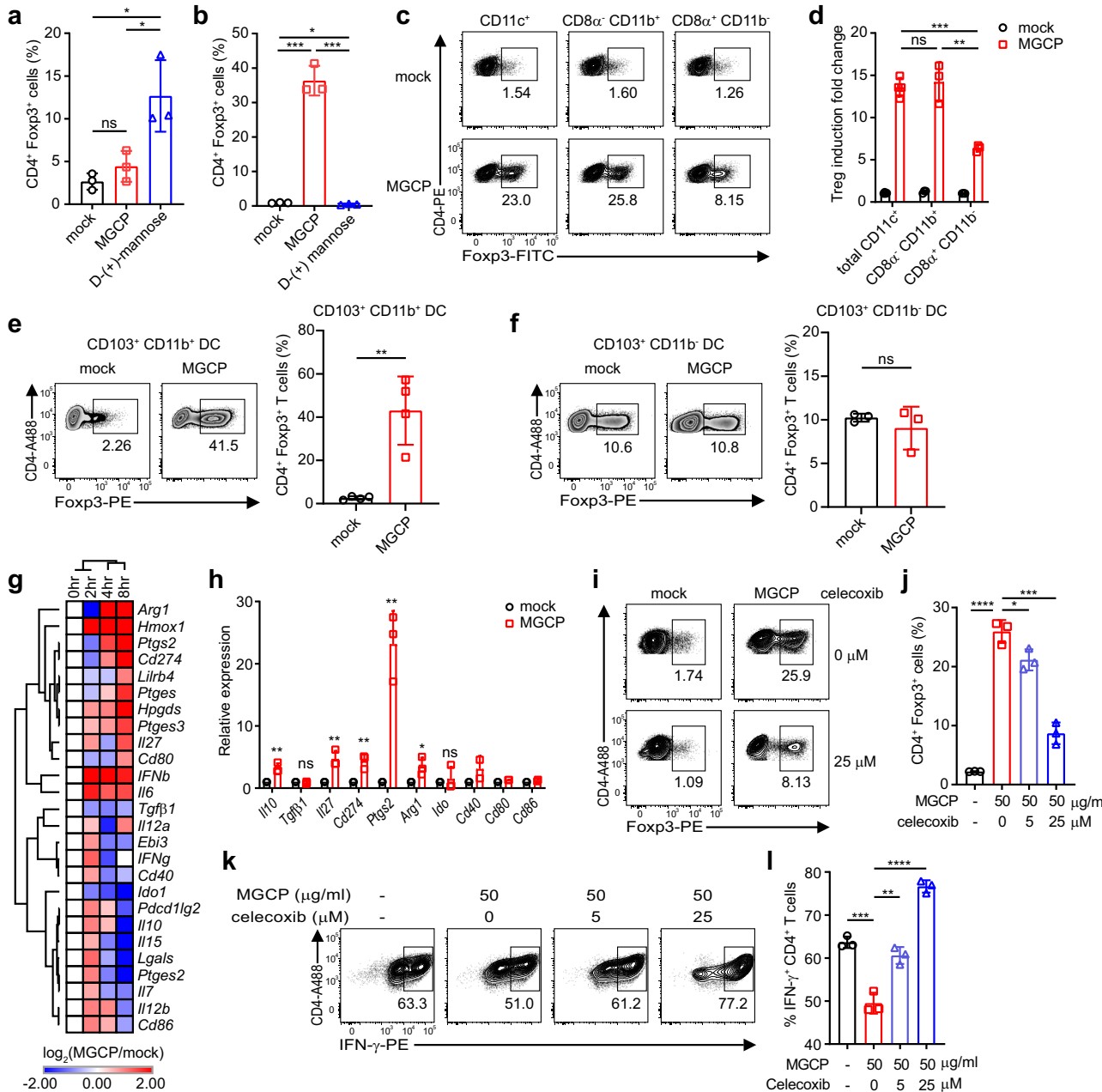

**Fig. 6 MGCP-mediated immune regulation is dependent on Cox2. a** Naive CD4$^+$ T cells were cultured with either MGCP or D-($+$)-mannose for 72 h in the presence of plate-coated anti-CD3 (1.5 μg/ml), soluble anti-CD28 (1.5 μg/ml), IL-2 (100 U/ml) and TGF-β1 (2 ng/ml) and frequency of Treg cells induced were analyzed. Result is representative data with 3 samples over 2 independent experiments. **b** MGCP or D-($+$)-mannose primed splenic DCs were cultured with naive CD4$^+$ T cells under suboptimal Treg-inducing condition and frequency of Treg cells induced were analyzed. Result is representative data of 3 samples over 2 independent experiments. **c–f** Indicated DC subsets (MHCII$^+$CD11c$^+$) were FACS sorted from spleen and small intestinal lamina propria (siLP), based on CD8 and CD11b (spleen) or CD103 and CD11b (siLP) expression, treated with MGCP or mock, and tested for their capability of inducing Treg cells under suboptimal iTreg inducing conditions. Representative FACS plots and relative frequencies of Treg cells induced by splenic DCs (**c**, **d**), by CD103$^+$CD11b$^+$ DCs (**e**) and by CD103$^+$CD11b$^-$ DCs (**f**) from siLP are shown. Result is pooled data of minimum 3 samples over 2 independent experiments (**d**), 4 samples over 2 independent experiments (**e**), and 3 samples over 2 independent experiments (**f**). **g** Total transcripts were isolated from splenic CD8α$^-$CD11b$^+$ DCs that were mock-treated or treated with MGCP for indicated time periods in four independent experiments, and assessed through RNA-seq analyses. Fold change of representative genes are shown. **h** Real-time PCR confirmation of genes associated with regulatory phenotype of splenic CD8α$^-$CD11b$^+$ DCs stimulated with mock or MGCP. Data pooled from three individual experiments. Data are analyzed with 3 samples over 3 independent experiments (Il10, Tgfβ1, Il27, Cd274, Ptgs2, Arg1, Ido) or 2 samples over 2 individual experiments (Cd40, Cd80, Cd86). **i–l** Splenic DCs pre-treated with MGCP were cultured with naive CD4$^+$ T cells in the presence of suboptimal amount of Treg or Th1 inducing cytokines. A selective inhibitor of Cox2, Celecoxib, was added in both DC stimulation and T-cell differentiation conditions. Representative flow cytometric plots (**i**) and percentage (**j**) of Treg cells induced by MGCP are shown. Result is representative data with 3 samples over 2 independent experiments. Representative flow cytometric plots (**k**) and frequency (**l**) of CD4$^+$ IFN-γ$^+$ T cells in the presence of Th1 skewing cytokines are shown. Data are analyzed from pooled data with three independent experiments. All bar graphs show the mean ± SD. *$p < 0.05$, **$p < 0.01$, ***$p < 0.001$, ****$p < 0.0001$ (two-tailed student's $t$ test). ns: not significant. Source data are provided as a Source data file.

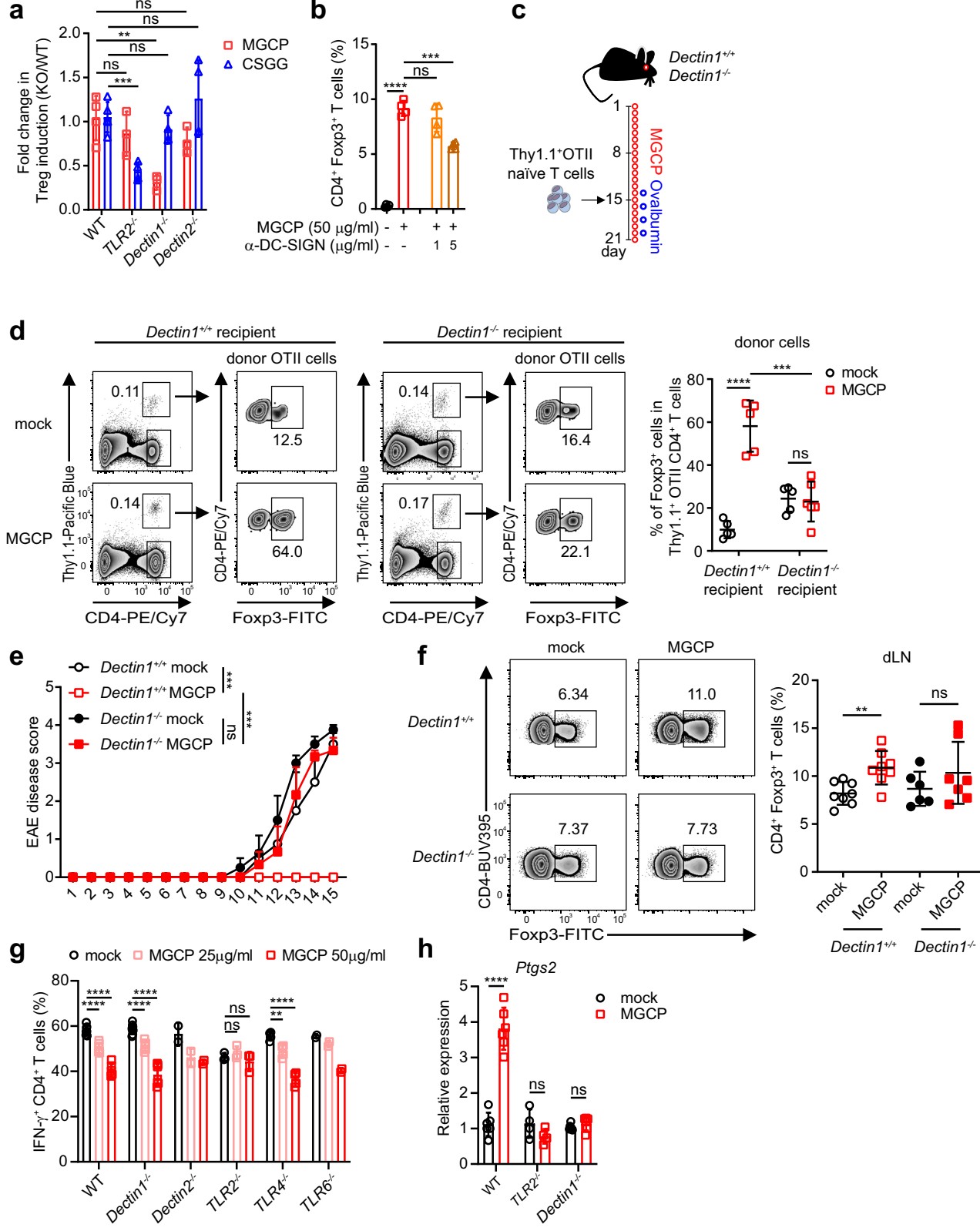

We next asked whether MGCP-mediated suppression of Th1 cell differentiation is also a DC-mediated Dectin1-dependent process. Surprisingly, when DCs isolated from mouse strains harboring deletion of different TLRs, were treated with MGCP and subjected to Th1 polarization assay, Dectin1 KO DCs could promote a characteristic reduction in Th1 cell induction, similar to that by WT DCs. On the other hand, $TLR2^{-/-}$ DCs displayed

complete resistance to this activity (Fig. 7g). Taken together these results suggested that MGCP-mediated functional modifications of DCs in order to promote immune tolerance are largely mediated by two independent PRR pathways. iTreg induction via Dectin1 and Th1 suppression via TLR2.

Despite such differential requirements of Dectin1 and TLR2, in our earlier experiments we found that pretreatment with the

**Fig. 7 MGCP associated PRR requirements in DCs. a** Splenic DCs isolated from indicated mice were treated with MGCP or CSGG followed by culture with naive CD4$^+$ T cells under suboptimal Treg-inducing conditions. Relative capacities of Treg cell induction are shown. Data normalized with Treg cell induction by WT DCs. Data are analyzed from 2 (WT MGCP, $TLR2^{-/-}$ MGCP, $Dectin1^{-/-}$ MGCP, $Dectin2^{-/-}$ MGCP, $TLR2^{-/-}$ CSGG, $Dectin1^{-/-}$ CSGG, $Dectin2^{-/-}$ CSGG) or 3 (WT CSGG) independent experiments with minimum 3 samples. **b** Splenic DCs treated with DC-SIGN antagonistic antibody prior to MGCP stimulation were cultured with naive CD4$^+$ T cells under suboptimal Treg skewing environment. CD4$^+$Foxp3$^+$ Treg percentage was analyzed. Data are analyzed from pooled data of 2 independent experiments with 4 samples. **c** Experimental scheme. Dectin1 sufficient and deficient mice were supplemented with mock or MGCP every day for two weeks prior to adoptive transfer of naive CD4$^+$ T cells from Thy1.1$^+$Foxp3$^{GFP}$ OTII mice. Recipient mice were continuously administered MGCP daily for an additional week and ovalbumin was fed every alternate days starting one day prior to adoptive transfer of the cells. **d** Representative flow cytometer plots and frequency of ovalbumin reactive Treg cells derived from donor cells was assessed in small intestine of recipient mice. Data are analyzed from 2 independent experiments with minimum 5 mice. **e, f** EAE was induced in WT and $Dectin1^{-/-}$ mice upon immunization with MOG$_{35-55}$ and complete Freund's adjuvant (CFA). Immunized mice were injected daily with MGCP, or distilled water as mock, from the day of immunization. **e** Graphical representation of EAE clinical scores for each treatment groups. Data are representative of 2 independent experiments and analyzed from minimum 3 mice. **f** Representative flow cytometric plots (left) and frequency (right) of CD4$^+$Foxp3$^+$ T cells in draining lymph nodes of indicated groups. Data are analyzed from pooled data of 2 independent experiments with minimum 6 mice. **g** DCs were isolated from indicated PRR knock-out mice and stimulated with MGCP followed by culture with naive CD4$^+$ T cells under suboptimal Th1 skewing conditions. Frequencies of Th1 cells generated are shown. Data are analyzed from 2 independent experiments with minimum 3 samples (WT, $Dectin1^{-/-}$, $TLR2^{-/-}$, $TLR4^{-/-}$) or 2 samples for $Dectin2^{-/-}$, $TLR6^{-/-}$ cells. **h** DCs were isolated from WT, $TLR2^{-/-}$, and $Dectin1^{-/-}$ mice and stimulated with MGCP for 8 h. Total mRNA was isolated and expression of Ptgs2, a gene that encodes Cox2, was assessed by qRT-PCR analysis. Data are analyzed from 2 independent experiments with minimum 4 samples. Each dot represents an individual mouse or sample. The graphs show the mean ± SEM. **e** ***$p < 0.001$ and ±SD. **a**, **b**, **d**, **f–h** **$p < 0.01$, ***$p < 0.001$, ****$p < 0.0001$ (two-tailed student's $t$ test). ns: not significant. Source data are provided as a Source data file.

Cox2 inhibitor is able to inhibit both activities of MGCP (Fig. 6i–l), suggesting that at least one molecular determinant, which is functionally affected downstream of both pathways, is MGCP-dependent expression of Cox2. We therefore assessed expression of Cox2 and other tolerogenic genes in wild-type, $Dectin1^{-/-}$, and $TLR2^{-/-}$ DCs after MGCP treatment. Indeed, whereas wild-type DCs exhibited increased expression of Ptgs2 following MGCP stimulation, this was significantly compromised in both $Dectin1^{-/-}$ as well as $TLR2^{-/-}$ DCs (Fig. 7h). In addition, MGCP-mediated enhanced expression of all the tested representative tolerogenic genes displayed complete dependence on TLR2. On the other hand, IL-10 and CD274 were found to be only partially dependent on Dectin1 signaling (Fig. S9b).

## Discussion

A number of studies have described the immunomodulatory properties of yeast polysaccharides and several polysaccharides have been pursued in clinics as adjuvants[7,8,41]. As a surrogate of yeast, zymosan is widely used for studying immunological properties of fungal polysaccharides. While most of these reports suggest a pro-inflammatory role of zymosan on the immune system[21,22], a handful of studies reported its immune suppressive potential as well[24,25]. Despite such advances, a comprehensive understanding of the contributions of individual yeast cell wall components on immune system is still missing. In this study, by identifying immunomodulatory components of the yeast cell wall and by evaluating their functional contributions toward pro- and anti-inflammatory immune responses, we significantly advance our basic understanding on the mechanisms by which the immune system communicates with resident microbiome, especially with commensal fungi. Furthermore, by identifying yeast cell wall-derived MGCP as potent immunomodulatory components that are effective against multiple experimental autoimmune diseases, we extend these findings toward potential therapeutic relevance.

Consistent with previous reports, our data revealed that zymosan promotes anti-cancer immune responses through inducing IFN-γ producing Th1 cell differentiation. We found that β-1,3-glucan of zymosan is essential for zymosan-mediated Th1 immune responses. More interestingly, we found that enzymatic removal of β-1,3-glucan from zymosan, promoted Treg cell differentiation in vitro and dampened anti-cancer immunity in vivo. These findings suggest that opposing effects of individual cell

surface polysaccharides shape up the immune-regulatory properties of zymosan in a combinatorial manner. While the pro-inflammatory role of zymosan is mediated by β-1,3-glucan, upon its removal, other polysaccharide components of zymosan oppose this function. Indeed, upon extensive purification of polysaccharides from yeast cell wall using a purification protocol that relied upon extraction of water-soluble polysaccharides and removal of insoluble β-1,3-glucan, we identified MGCP, which was found to be effective in promoting the induction of iTreg cells, as well as repressing pro-inflammatory Th1 cell differentiation in vitro. One outstanding issue with regard to MGCP-mediated repression of EAE in vivo that remains less resolved, however, is that, MGCP treatment results in increase in Treg cell frequency in draining lymph nodes of experimental animals, as well as suppresses Th1, Th17, and IFN-γ$^+$IL-17$^+$ double-positive population. It seems likely that multiple mechanisms involving MGCP-dependent Treg-mediated suppression, in conjugation with its effect on effector T-cell differentiation, are responsible for the observed anti-inflammatory outcome.

Our effort to dissect the identity and structural specifications of the key immunoregulatory component of MGCP revealed important mechanistic issues that further highlight the novelty of this study. Glutaraldehyde-polymerized allergoids conjugated to nonoxidized mannan has been previously shown to have Treg-inducing properties through a program death-ligand 1 (PD-L1)-mediated mechanism[16]. However, this appears to be not the key Treg-inducing pathway in the in vitro and in vivo experimental conditions employed in this study. Furthermore, D-(+)-mannose, the monomeric building block of mannan, has also been demonstrated to have Treg-inducing property by acting directly on naive T cells in a DC-independent manner[33]. Therefore it seemed possible that monomeric D-(+)-mannose produced after degradation of mannan in MGCP may be instrumental in promoting iTreg induction. Our results demonstrating that the Treg-inducing property of MGCP is a DC-dependent phenomenon, strongly suggests this is unlikely to be the case.

Our analyses showed that mannan is the abundant component of purified MGCP, in which β-1,6-glucan is 15% by weight as calculated by monosaccharide composition. However, liquid chromatography with gel filtration, as well as enzymatic depletion experiments followed by functional assays, revealed that even though the glucan is a minor component, it is essential for promoting the immunoregulatory properties of MGCP. The

relatively less abundant polysaccharide with β-1,6-glucan backbone, with a side chain of a single β-1,3-linked glucose, was found to be the key immunoregulatory component in MGCP. The β-1,3-linked glucose side chain appears to add important functional specificities to MGCP, because it distinguishes immunoregulatory properties of MGCP from that of CSGG, the cell surface β-glucan/galactan, which we recently demonstrated to be potent Treg-inducing components of *B. bifidum*[28]. While both MGCP and CSGG could promote iTreg induction, only MGCP could restrict Th1 cell generation under Th1 skewing conditions.

Substantiating this notion, we found that unlike CSGG, which promotes iTreg induction through a TLR2-dependent mechanism[28], MGCP employs a Dectin1-dependent signaling axis in DCs to exert its effect. Dectin1 is a well-studied receptor for fungal recognition, which is primarily known to induce pro-inflammatory immune responses like Th17 differentiation[42]. Therefore, its anti-inflammatory property observed in our study appears to be a disparate function of the receptor in the context of MGCP-mediated immune regulation. Such context-dependent distinctive functions of Dectin1 may arise from differential binding affinity with its ligands, as well as combinatorial downstream signaling resulting from its already documented parallel engagement of additional TLR pathways[43]. Further underscoring the complexity of mechanisms involved, we found that unlike its iTreg induction activity, MGCP-mediated repression of Th1 cell differentiation is dependent on TLR2 expression on DCs, even though both pathways are functionally dependent on MGCP-mediated upregulation of Cox2. It seems likely that subtle structural differences between otherwise similar polysaccharides derived from different species may exert distinct effects on host immune system through variable binding preferences and downstream signaling events mediated by different PRRs. In the future, identification of more such functionally relevant commensal-derived polysaccharide followed by an in-depth understanding of the structural and molecular basis of their functions may prove extremely beneficial in order to employ this knowledge against autoimmune and allergic diseases.

## Methods

**Mice.** Mice were maintained in the animal facility POSTECH Biotech Center in specific pathogen-free condition. The POSTECH Institutional Animal Care and Use Committee approved all the experiments. C57BL/6 mice were maintained and inbred in POSTECH. *Foxp3-eGFP* and *Tlr2*[−/−] mice were obtained from the Jackson Laboratory. Thy1.1[+]*Rag1*[−/−]TCR Vα2[+] Foxp3[EGFP] OTII (*Rag1*[−/−]OTII TCR transgenic) and *Rag1*[−/−] mice were obtained from Taconic. *Dectin1*[−/−] and *Dectin2*[−/−] animals were kindly provided by Dr. Yoichiro Iwakura (Tokyo University of Science, Japan). Dr. Andrew Macpherson (Bern Univ., Switzerland) and Dr. David Artis (Then at Univ. Pennsylvania, currently at Cornell Univ., USA) kindly provided GF C57BL/6 mice. GF mice were maintained in sterile flexible film isolators (Class Biological Clean Ltd., USA). *CBir* mice were provided from Charles O. Elson, University of Alabama at Birmingham. 6–12-week-old sex and age-matched mice were utilized for all the experiments.

**Purification of Mannan/β-glucan-containing polysaccharides from yeast cell wall.** Twenty grams of yeast extract (BD Biosciences), 2 g of polysorbate 80 (Sigma-Aldrich), 4 g of ammonium citrate (Sigma-Aldrich), 10 g of sodium acetate (Sigma-Aldrich), 0.2 g of magnesium sulfate (Sigma-Aldrich), 0.1 g of manganese sulfate (Sigma-Aldrich), and 4 g of dipotassium phosphate (Sigma-Aldrich) were dissolved in 2 L of distilled water. The solution was autoclaved and cooled at room temperature. Trichloroacetic acid (TCA, Sigma-Aldrich) was added to the solution for final concentration 0.4% and incubated at 4 °C with magnetic stirring overnight. The supernatant of the TCA-treated solution was incubated with three volumes of chilled ethanol at −20 °C overnight. The solution was centrifuged, supernatant discarded, and pellet was dried and suspended in 10 mM Tris buffer including 20 mM MgCl₂, 20 mM CaCl₂ (pH 7.5). Suspended solution was treated with RNase (Sigma-Aldrich) to final concentration 0.4 mg/ml and DNase (Roche) and incubated at 37 °C overnight. The solution was treated with sodium azide to 0.05% final concentration and incubated for 30 min at 37 °C. After incubation, Pronase (protease from *Streptomyces griseus*, Sigma-Aldrich) solution was added to a concentration of 0.3 mg/ml and incubated at 37 °C overnight. Following this, Pronase was added once more for 0.3 mg/ml final concentration and incubated for

additional 2 h. TCA was added to this solution to a final concentration of 0.4%, incubated at 37 °C for 2 h, centrifuged, ethanol precipitated and the pellet suspended in 100 mM Tris buffer (pH 7.5). Equal volume of phenol was added to the solution, mixed well by inverting, and centrifuged. The upper layer was transferred to the new tube and subjected to phenol extraction once again. The upper layer was transferred to a new tube and extracted twice with equal volume of iso-amyl alcohol and chloroform solution, 1:29 (v:v). Polysaccharides from the upper layer were dialyzed against distilled water for 3 days and freeze-dried. Concentration of polysaccharides was measured by acid phenol assay[44]. For purifying polysaccharides from yeast cell wall after removal of β-1,3-glucan, zymolase was added to the solution of yeast extract to eliminate β-1,3-glucan by following manufacturer's protocol. Hereafter, same protocol was performed to isolate polysaccharides.

**Polysaccharide composition analysis.** Monosaccharide composition and linkage analysis were performed by transforming the sample in the corresponding acetylated methylglycoside derivatives or in the corresponding partially methylated and acetylated alditols, respectively, as previously described[45]. Derivatives were analyzed with gas chromatographer equipped with an electron impact mass spectrometer detector (GC-MS, Agilent GC 6850/5973), the SPB-5 capillary column (Supelco, 30 m × 0.25 i.d.) was run under 0.8 ml/min flow rate with Helium as carrier gas, with temperature program: 150 °C for 5 min, 150 up to 300 °C at 10 °C/min, 300 °C for 12 min. Electron impact mass spectra were recorded with an ionization energy of 70 eV and an ionizing current of 0.2 mA. The derivatives were recognized by comparing their fragmentation pattern and their retention time with that of standards prepared from commercial monosaccharides.

**Molecular weight analysis and chromatographic purification.** Purified polysaccharides from yeast cell wall were analyzed by high-performance liquid chromatography (HPLC) to determine their molecular weight or to fraction them according to molecular weight. For separation, Polysaccharides were dissolved into distilled water and filtered with 0.45-m syringe filter prior to fractionation by size exclusion chromatography (TSKgel G5000PW_XL) mounted on an HPLC system (Gilson), run by using distilled water was used as eluent under 0.3 ml/min flow rate; chromatogram profile was monitored with refractive index detector (Knauer K-2301) and UV detector at wave length 206 nm. For examining the molecular weight, polysaccharides were dissolved in 50 mM of NH₄HCO₃ buffer and subjected to HPLC analysis (Agilent 1100, equipped with both UV and refractive index detector) by size exclusion chromatography on the same column (TSKgel G5000PW_XL), ran at 0.8 ml/min flow rate and using 50 mM of NH₄HCO₃ as eluent. Dextran standards (1, 5, 50, 150, 410, 670 kDa) were used to build the calibration curve used to evaluate the MW of the sample.

**Nuclear magnetic resonance (NMR) analysis.** The structures of polysaccharides in the yeast extract were assessed through NMR using the same methods as described previously[28,46]. In brief, the sample (ca. 5 mg) was dissolved in D₂O and measured by using a DRX 600 MHz fitted with cryogenic probe (Bruker). The full set of 2D spectra (COSY, TOCSY, NOESY, HSQC, HSQC-TOCSY, and HMBC) were measured at 37 °C (310 K).

Homonuclear experiments were recorded using 512 FIDs of 2048 complex data points, setting 24 scans per FID for all experiments. Mixing time of 100 and 200 ms was applied for TOCSY (and HSQC-TOCSY, as well) and NOESY, respectively. ¹H–¹³C heteronuclear experiments were acquired with 512 FIDs of 2048 complex points with 40 scans for the HSQC, while 80–100 scans were used for the others. Standard Bruker software, Topspin 3.1, was used to process and analyze all spectra.

Characterization of the NMR spectra relied on established approaches. In brief, the identification of all proton spin systems was achieved by tracing the spin connectivities derived from the double quantum filtered correlated spectroscopy (DQF-COSY) and the total correlation spectroscopy (TOCSY) spectra. For the glucose units, the anomeric proton (H-1) had a set of correlations reaching the two H-6 protons whereas for the mannose residues, the scalar coupling in the TOCSY spectrum stopped at H-2, and the finding of the other protons resumed by analyzing the correlations starting from position H-2.

The identification of the carbon chemical shifts of each residue was attained by combing the information of the heteronuclear spectra, ¹H,¹³C HSQC, and ¹H,¹³C HSQC-TOCSY spectra, and counterchecked with the information from the ¹H,¹³C HMBC spectrum. Finally, the merging of data from NOESY, and HMBC spectra, allowed the establishment of the complete polysaccharide structures by detecting the *inter*-residue connectivities.

**Isolation of lymphocytes and flow cytometry analysis.** Naive CD4[+] T cells were isolated from pLN, mLN and spleens by using FACS sorter (Astrios, Beckman Coulter) or EasySep™ Mouse naive CD4[+] T-cell Isolation Kit (STEMCELL Technology) following manufacturer's protocol. For isolation of lymphocytes from colon, small intestine and spinal cord, tissues were opened longitudinally and rinsed with PBS to remove mucus and feces. Intestines were cut into pieces with 0.5–1 cm and incubated with PBS containing 10 mM EDTA, 20 mM HEPES, 1 mM sodium pyruvate, and 3% of FBS while stirring with magnetic bar for 20 min at 37 °C. Tissue was washed with PBS, then minced followed by incubating in RPMI

1640 media with 3% FBS, 20 mM HEPES, 1 mM sodium pyruvate, 0.5 mg/ml of Collageanse D (Roche), and DNase I (Sigma-Aldrich) for 45 min at 37 °C. Tissue was incubated for additional 5 min in the presence of 10 mM EDTA. Supernatant was filtered with 100 μm cell strainer and transferred into chilled PBS to remove remaining enzymes and EDTA. Cells were loaded onto 40 and 75% Percoll™ (GE Healthcare) gradient. Lymphocytes were harvested from interface percoll gradient layer and washed with DMEM media supplemented with 1% FBS and 1% penicillin/streptomycin. For analysis of cytokine, cells were stimulated with PMA (Calbiochem) and Ionomycin (Calbiochem) in the presence of Golgistop (BD Biosciences) for 4–5 h in complete RPMI media containing 10% FBS, 1% penicillin/ streptomycin, 2 mM L-glutamine, 1 mM sodium pyruvate, non-essential amino acids, and 0.1% β-ME (v/v) at 37 °C. Cells were stained for analysis with flow cytometry following manufacturer's protocol. For cytokine analysis, total lymphocytes were stimulated with 50 ng/ml PMA and 1 μM ionomycin in the presence of GolgiStop (BD Biosciences) for 5 h. Cells were subjected to intracellular staining by using Intracellular Fixation & Permeabilization Buffer Set (eBioscience) following manufacturer's protocol. Following reagents were used for cell staining. Live/DEAD fixable viable dye (Life Technologies), Fixation/Permeabilization buffer (eBioscience), Permeabilization buffer (eBioscience), IC fixation buffer (eBioscience), and antibodies. Following antibody clones were used: CD4 (RM4-5), CD44 (IM7), CD62L (MEL-14), CD45.1 (A20), CD103 (2E7), Foxp3 (FJK-16s), CTLA-4 (UC10-4B9), Nrp1 (3DS304M), IL-10 (JES5-16E3), IFN-γ (XMG1.2), IL-17A (TC11-18H10.1), CD11c (N418), CD11b (M1/70), F4/80 (BM8), MHCII (M5/114.15.2). A comprehensive list with detailed information on the source, clone, format, and Catalog number of all the Antibodies are provided in Table S2. The gating strategies used for analyses of FACS data are presented in Figs. S10, S11. FACS data were acquired using BD FACSDiva software and analyzed using the software Flowjo (Treestar) v10.5.0.

**Antigen-presenting cell-dependent in vitro T-cell differentiation**. For most in vitro experiments, DCs were isolated by CD11c magnetic beads (Miltenyi Biotech) following manufacturer's protocol. In a few experiments, cells were isolated from spleen (MHCII+CD11c+; MHCII+CD11c+CD11b+CD8α−; MHCII+CD11c+CD11b−CD8α+) or small intestine (MHCII+CD11c+CD11b+CD103+F4/80−; MHCII+CD11c+CD11b−CD103+F4/80−; MHCII+CD11c+CD11b+CD103−F4/80+) by FACS sorting or from colon by CD11c magnetic beads (Miltenyi Biotech). Isolated $2 \times 10^4$ cells of DCs were treated with indicated polysaccharides or MGCP and cultured in complete RPMI 1640 media for 14 h in the presence of 10 ng/ml GM-CSF (Perprotech). For stimulation of DCs with multiple polysaccharides, DCs were stimulated with 50 μg/ml of MGCP in most of the in vitro experiments except a few cases, where MGCP concentrations are mentioned or indicated polysaccharides. Stimulated DCs were washed and cultured with $2 \times 10^5$ naive CD4+ T cells for 3 days. Experiments were performed under suboptimal CD4 T-cell skewing condition. Following is each subset of CD4 T-cell differentiating condition. Th0: 0.1 μg/ml of anti-CD3 (BioXcell), 100 U/ml of IL-2, 10 ng/ml GM-CSF, and 10 μg/ml of anti-IL-4 (BioXcell); Th1: 0.1 μg/ml of anti-CD3, 100 U/ml of IL-2, 2.5 ng/ml of IL-12, 10 ng/ml of GM-CSF, and 10 μg/ml of anti-IL-4; Th17: 0.1 μg/ml of anti-CD3, 1 ng/ml of IL-6, 0.1 ng/ml TGF-β1 (Miltenyi Biotech), 10 ng/ml of GM-CSF, 10 μg/ml of anti-IFN-γ (BioXcell), and 10 μg/ml of anti-IL-4; Treg: 0.1 μg/ml of anti-CD3, 100 U/ml of IL-2, 0.1 ng/ml TGF-β1, and 10 ng/ml GM-CSF. GM-CSF was included in culture media to enhance DC viability. Cells were analyzed with flow cytometry after 3 days of culture. For co-culture experiments utilizing antagonists, antagonists were treated to splenic DCs before MGCP stimulation. For experimental procedures using polysaccharide-cleaving enzymes, polysaccharides were digested with indicated enzymes following manufacturer's protocol prior to treatment of splenic DCs and cultured with naive CD4+ T cells. For experiments with Celecoxib (Sigma-Aldrich), splenic DCs incubated with MGCP, were treated for 30 min with Celecoxib, then cultured with naive CD4+ T cells in the presence of celecoxib. Following reagents were used: pustulanase (Prokazyme), zymolyase (MPbio), anti-Mincle mAb (MBL), anti-DC-SIGN Ab (Abcam), anti-human CD206 (MMR) mAb (Biolegend), Zymosan (Sigma-Aldrich), Zymosan Depleted (InvivoGen), D-(+)-mannose (Sigma-Aldrich). For cytokine analysis after in vitro differentiation, cells were restimulated with 50 ng/ml PMA and 1 μM ionomycin in the presence of GolgiStop (BD Biosciences) for 5 h. Cells were subjected to intracellular staining by using Intracellular Fixation & Permeabilization Buffer Set (eBioscience) following manufacturer's protocol. Following antibodies were used: Live/DEAD fixable viable dye (Life Technologies), CD4 (RM4-5), CD45 (30-F11), Foxp3 (FJK-16s), IFN-γ (XMG1.2), IL-17A (TC11-18H10.1).

**B16.F10 melanoma engraftment**. B16.F10 melanoma cell line was purchased from ATCC. B16.F10 melanoma cells were grown in DMEM supplemented with 10% FBS. B16.F10 cells ($1 \times 10^5$ cells in 100 μl PBS) were injected subcutaneously and tumor growth was monitored over time. Tumor volume was calculated by following formula (major circumference × minor circumference$^2$)/2. Tumor-bearing mice were treated with zymosan (200 μg) or β-1,3-glucan removed zymosan through intraperitoneal route every alternate day. β-1,3-Glucan was removed from the zymosan by using zymolase enzyme following by manufacturer's protocol.

**Induction of experimental colitis through transfer of CD4+ T cells**. Experimental colitis was induced by following previously reported method[29]. In brief, FACS sorted CD4+Foxp3−CD44loCD62Lhi naive T cells ($1 \times 10^6$) from either congenic CD45.1+ Foxp3-EGFP or CBir mice were transferred to Rag1 deficient mice. To assess efficacy of in vitro generated MGCP-induced Treg cells, naive CD4+ T cells were adoptively transferred together with indicated Treg cells ($2 \times 10^5$ cells) simultaneously. For induction of colitis with naive CD4+ T cells from CBir mice, recipients were orally administered with mock or MGCP every other day during entire experimental period. Progression of colitis was monitored by measuring body weight two times a week. Mice were sacrificed when their body weight decreased to 20% of initial weight. Disease severity was analyzed by measuring colon length, histological assessment, and cytokine production from donor naive CD4+ T cells.

**In vivo adoptive transfer experiments**. 200 μg of MGCP was orally administered to recipient WT or Dectin1−/− mice for 2 weeks every day before transfer of naive CD4+ T cells. Sorted CD4+Foxp3−CD44loCD62Lhi naive T cells (purity for >99%, 1.5–2 × 10⁶) from RAG1−/−Thy1.1+Foxp3EGFPOTII mice or CD45.1+ Foxp3-EGFP mice were transferred intravenously to these MGCP administered recipient mice. Mice were treated with MGCP every day for 1 more week before analyses. For experiments with transfer of OTII naive CD4 T cells, MGCP were treated daily along with 20 mg of OVA protein every alternate day for 1 more week.

**Induction of experimental autoimmune encephalomyelitis (EAE)**. Mice were immunized with 200 μg of synthesized MOG$_{35-55}$ peptide (Anygen, Korea) in CFA (BD Difco) emulsion containing 400 μg of Mycobacterium tuberculosis (BD Difco) on day 0. Mixture of MOG$_{35-55}$ peptide, CFA, and M. tuberculosis were subcutaneously injected into both right and left flank side of the mice. Mice were administered 400 ng of pertussis toxin (List Biological Laboratories Inc.) through intraperitoneal route on days 0 and 2. Clinical symptom was monitored every day. Disease score was measured according to the following score: 0, no sign of disease; 1, limp tail or hind weakness; 2, partial hind limb paralysis or uncoordinated movement; 3, complete hind limb paralysis; 4, hind limb paralysis and both forelimbs paralysis; 5, moribund or dead. From day 0, immunized mice were injected intraperitoneally with mock (DW) or MGCP (200 μg) every day until the end of experiment.

**RNA sequencing**. CD8−CD11b+ DCs were FACs sorted from spleen and treated with mock (DW) or MGCP (25 g) for 2, 4, and 8 h. DCs were harvested and dissolved in TRIzol reagent. Total transcripts were purified by using RNA isolation kit (GeneAll Biotechnology). Ribospin TMII (GeneAll Biotechnology) was used for isolation of total RNA. Library was prepared through TruSeq Stranded mRNA sample Preparation Kit (Illumina, San Diego, CA). RNA-sequencing analysis was performed with NextSeq 500 Sequencing platform. For transcriptome analysis in colonic DCs in vivo, GF mice were administered mock and MGCP every day for 2 weeks. Colonic CD11c+ DCs from mock and MGCP supplemented GF mice were isolated with microbeads following manufacturer's protocol from total colonic lamina propria cells. Total RNA was purified from the colonic DCs of mock or MGCP fed mice by using Ribospin TMII (GeneAll Biotechnology). TruSeq Stranded mRNA sample Preparation Kit (Illumina, San Diego, CA) was utilized for preparing library. Transcriptome analysis was performed with NextSeq 500 Sequencing platform. The RNA-seq data were deposited in the Gene Expression Omnibus (NCBI) data repository under accession number GEO: RNA-seq data: GSE126937.

**Bacterial 16s rDNA sequencing**. A total of 24 mouse fecal samples for three groups (HC, MOCK, and MGCP) at two different time points (D0 and D32) were processed for DNA extraction using a DNeasy PowerSoil Kit (Qiagen) following manufacturer's instructions. Samples were further processed for amplicon (V3-V4 region). Microbiome library was generated with Herculase II Fusion DNA Polymerase Nextera XT Index Kit V2, followed by metagenomic sequencing on an Illumina Miseq platform. Raw sequencing data were processed using QIIME2 v2020.8.0 tool[47]. DADA2 was used for trimming filtering and denoising[48]. Samples were rarefied to a depth of 5485 sequences per sample prior to alpha diversity (Shannon entropy) and beta-diversity (Bray–Curtis index) calculation. Pairwise permutational multivariate analysis of variance (PERMANOVA) was used to test significance of differences in beta-diversity in each pair of samples. QIIME2 tool RESCRIPt was used for processing and filtering SILVA 138.1 database for taxonomy assignment[49,50]. SILVA sequences were trimmed using primers for the V3-V4 region. Taxonomic classification was performed using sklearn[51].

**Quantitative reverse transcriptase PCR**. Splenic CD8α−CD11b+ DCs or total CD11c+ DCs were isolated and by FACS sorting or magnetic beads and stimulated with mock (DW) or MGCP (25 μg) for 8 h. Cells were harvested and dissolved in TRIzol reagent. Total transcripts were purified following manufacturer's protocol. cDNA was synthesized from purified mRNAs by using M-MLV reverse transcriptase (Promega). For examination of total transcriptome of colonic CD11c+ DCs from mock and MGCP supplemented GF mice, GF mice were fed with mock or MGCP (200 μg) for 2 weeks every day. Colonic CD11c+ cells were sorted with microbead (MACs) and dissolved in TRIzol reagent. Same protocol with

transcriptome isolation from splenic CD8α⁻CD11b⁺ DCs was performed for purifying total RNA and synthesis of cDNA. Expression level of indicated markers was analyzed with prepared cDNA and following primer pairs: HPRT, (forward) 5′-TTA TGG ACA GGA CTG AAA GAC-3′ and (reverse) 5′-GCT TTA ATG TAA TCC AGC AGG T-3; IL-10, (forward) 5′-ATA ACT GCA CCC ACT TCC CA-3′ and (reverse) 5′-TCA TTT CCG ATA AGG CTT GG-3′; TGF-β, (forward) 5′-CTC CCG TGG CTT CTA GTG C-3′ and (reverse) 5′-GCC TTA GTT TGG ACA GGA TCT G-3′; PD-L1, (forward) 5′-GCT CCA AAG GAC TTG TAC GTG-3′ and (reverse) 5′-TGA TCT GAA GGG CAG CAT TTC-3′; IDO, (forward) 5′-GCT TTG CTC TAC CAC ATC CAC-3′ and (reverse) 5′-CAG GCG CTG TAA CCT GTG T-3′; COX2, (forward) 5′-TGG CTG CAG AAT TGA AAG CCC T-3′ and (reverse) 5′-AAA GGT GCT CGG CTT CCA GTA T-3′. All data were normalized to the expression level of hypoxanthine-guanine phosphoribosyl transferase (HPRT). Results were further analyzed as relative expression levels compared to expression levels of mock control. Primer sequence information is listed in Table S3.

**Histology.** One centimeter of colon or spinal cord was fixed in 10% formaldehyde and embedded in paraffin blocks. Paraffin blocks were sectioned at 3-μm thickness and stained with Hematoxylin (Sigma-Aldrich) and Eosin (Sigma-Aldrich). Colon histolopathology score was measured with three sections of each tissue per mouse at ×100 magnitude. Score was assessed as follows: 0, normal tissue; 1, cellular infiltration around crypt bases and minimal loss of goblet cells; 2, infiltrating cells in muscularis mucosa and moderate structural destruction of epithelial cells with elongated crypt and extensive loss of goblet cells; 3, severe inflammation marked with extensive cellular infiltration in muscularis mucosa and edema and tissue destruction with extensive loss of goblet cells and minimal loss of crypts; 4, severe cellular infiltration in submucosa and extensive loss of crypts.

**Statistical analysis.** Statistical analysis was performed with Graphpad Prism software (La Jolla, USA). Differences between control and experimental groups were evaluated using two-tailed unpaired-Student's t test. Data are presented as mean ± SD or ±SEM.

**Reporting summary.** Further information on research design is available in the Nature Research Reporting Summary linked to this article.

## Data availability
All data needed to evaluate the conclusions in the paper are present in the paper and/or the Supplementary Materials. Additional data related to this paper may be requested from the authors. RNA-seq data were deposited in the Gene Expression Omnibus (NCBI) data repository under accession number GEO: RNA-seq data with splenic DCs: GSE150685. RNA-seq data with colonic DCs: GSE126937. Source data are provided with this paper.

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

## Acknowledgements

We thank Y. Iwakura, Tokyo University, for kindly providing *Dectin1*$^{-/-}$ and *Dectin2*$^{-/-}$ mice. We also thank H. Jung for technical support for cell sorting and K.S. Lee for mouse husbandry. This work was supported by grant from Institute for Basic Sciences (project IBS-R005), Korean Ministry of Science and ICT (to S.-H.I. and D.R.), and the Bio & Medical Technology Development Program of the National Research Foundation (NRF) funded by the Korean government (MSIT) (No. NRF-2020M3A9G3080282; to K.S.K.).

## Author contributions

C.L., D.R., K.S.K., and S.-H.I. designed the experiments. C.L. performed most of the experiments. C.L., S.-H.I., and D.R. analyzed data. R.V., S.B., and E.-J.J. participated in performing experiments. E.-J.J. and S.B. performed polysaccharides purification from yeast cell wall. H.-J.K. and G.S. performed animal experiments. G.-C.K. and C.J.K. performed RNA-seq data analysis. C.D.C., I.S., and A.M. performed and analyzed NMR, composition and linkage analysis data. A.L. and S.P. analyzed 16s rDNA sequencing data. S.L. and D.S.H. performed HPLC and NMR analyses. C.D.C. and A.M. participated in performing HPLC and NMR analysis and composition analysis of polysaccharide. D.R. and S.-H.I. supervised, provided intellectual suggestions during the course of the study, and wrote the manuscript with C.L.

## Competing interests

H.-J.K. and G.S. are employees of HEM and ImmunoBiome, respectively, but declares no conflicts of interest for this paper. C.L., R.V., and S.-H.I. filed a patent application as major inventors. S.-H.I. is the CEO and major shareholder of ImmunoBiome, but declares no conflicts of interest for this paper. Other authors declare that they have no competing interests.
