## [Peer Review File · Nature Communications]

REVIEWER COMMENTS

Reviewer #1 (Remarks to the Author):

After analyzing the manuscript entitled "Commensal yeast cell surface polysaccharide activities unleashed upon removal of β -1,3-glucans restrain inflammatory disorders", which aimed to develop immunomodulatory activities of these polysaccharides after removal of β -glucan- (1,3) -linked by enzymatic action. It is an excellent research and can be published, but the part related to chemical structures must be revised, while the part of biological activity is well developed and with innovative results and must add to those of other research. Some suggestions and questions are presented below:

1. The Abstract is presented in a generally extremely discursive and little orientative way, therefore bringing some experimental results obtained, mainly the significant and comparative ones with respective percentages in several experiments that will bring greater orientation to the readers
2. Between lines 86 to 92 of the "Introduction section" bring precise information on the chemical structures of the polysaccharides present in the cell wall of this yeast and better report the biological action of zymosan in relation to the immune system
3. The extraction and purification process (materials and methods) was developed in this research, so the extraction process must be presented in the text, the presentation is incomplete and must be reorganized
4. Add some lines in the methodology (analysis of the composition of Polysaccharides) developed to obtain the composition of monosaccharides. The way it is presented brings many doubts to the understanding, mainly the process of methylglycosides and patterns used.
5. Present $^1\text{H-NMR}$ or $^{13}\text{C-NMR}$ spectra of the polysaccharides present in (a) purified yeast extract; (b) polysaccharide after removal of β -glucan by zymolase; (c) residual polysaccharide + zymosan. In this way, we will have a general view of the chemical structures obtained from the yeast extract. Thus, by integrating the anomeric peaks ($^1\text{H-NMR}$) of these structures, we will have an orientation of the type of units of the remaining polysaccharides. $^{13}\text{C-NMR}$ observe the anomeric signals and the linkage region.
6. I understand that "Figure S1. Chemical and functional properties of MGCP" must be present in the text of the manuscript and not as supplementary material. Here are some important results for understanding the changes related to the chemical structures presented here.
7. The results related to the different chemical structures should be better discussed and clarified in "Results" and "Discussion".
8. Observing the text of lines 387 to 392 regarding the polysaccharide MGCP is confused because when viewing Figure S1 (E) it shows a B-Glucana-(1,6)-linked with single ramification - (1,3)-linked, however the Figure S1 (a) and (c) show abundant values of mannose. Thus, it brings doubts and the text must be revised
9. The extraction and purification process should be improved and an extraction and purification scheme can be added in Figure S1. This scheme should clarify to readers the modifications developed in yeast polysaccharides
10. In Figure 2 yeast cell wall derived MGCP, in items (b) and (c) the information regarding the structures of MGCP are not correct, because they are not structures produced by NMR, these are

structures obtained from literature. In this manuscript the necessary NMR spectra were not presented and there are no experiments to confirm these chemical structures.

11. Bring a better discussion in "Results" and also in "Discussion" section related to Figure 1. Zymosan β -1,3-glucan. In an attempt to clarify the results observed with Zymosan and zymosan together with β -(1,3)-glucanase.

Reviewer #2 (Remarks to the Author):

The authors are addressing an interesting aspect of immune-modulatory properties of components of the yeast cell wall. They succeed in some respects especially the biochemistry, but some controls are missing and conclusions for the Treg/Th1 component is not strong and the Cox2 as a mechanism is insufficient. The mouse models show an effect but possibly not for the reasons they state.

1. Fig1. The authors should state in text the inclusion of GM-SCF (not clear why they would need this for polarisation of iTregs or Th1).

The conclusion that B 1,3 Glucan enhances Th1 and restrains iTregs is a bit misleading. In fact, one might argue that there's no effect on iTreg induction with their no treatment group, Th0 control is required, as their inducing conditions actually shows no Foxp3.

1b, Needs control. Th0, and preferably Th17 as the interplay with Tregs is at the center of this story. The effect of Th1 induction by B 1,3 Glucan at 200ug is modest but believable. However, in Fig 1d, to claim suppression here is inaccurate. The authors should show expression of Foxp3 in this figure (i.e is decreased percentage due to induction of Foxp3 or actual antagonism of Tbet programme). Controls of Th0 and Th17 would also support their claim.

2. Fig2. Shows nicely the IFN issue discussed above (see Fig2e. The x-axis, cell size moves closer to zero).

3. Fig3. Is not convincing. In general this comment goes to other autoimmune models in this paper. MGCP induction of de novo iTregs may be happening. However, because Tregs regulated antigen specific responses at a DC level, the reduced Th1 response observed is likely due to Lack of antigen (Ethan Shevak's Lab showed this, Nat Immunology 2019, Akkaya).

4. Fig4. This is consistent with their Data in Figure 4 (EAE model) which shows that there's no accumulation of iTregs in the CNS but rather in dLN and Fig4 I shows nicely the BOTH antigen-specific TH1 and Th17 and also the dual producers are both downregulated, making the claim about regulation of Th1 weak.

5. The authors should demonstrate that MGCP does not work during an inflammatory model in dectin-1 mice (eg: EAE)

6. The authors should show that MGCP is not present in depleted zymosan, which lacks TLR2 stimulatory activity (PMID: 12719479)

Reviewer #3 (Remarks to the Author):

The manuscript by Rudra et al. describes a soluble yeast cell wall preparation (MGCP) with immunomodulatory properties. Using in vitro and several in vivo models, the authors convincingly demonstrate the capacity of MGCP to induce Tregs by modulating DC function, and to suppress the generation of IFN- γ producing CD4⁺ cells. This correlated with significantly reduced disease severity in several models of inflammatory diseases. They furthermore showed that Dectin-1 and TLR2 are involved in the effect of MGCP on DCs and T cells.

These findings are novel and very interesting both from a biological perspective and due to the possible implications for novel therapeutic approaches.

The experiments are technically sound. There are however some points that need to be addressed:

1. The authors focused their in vivo analyses on CD4⁺ T cells– what about APCs, and myeloid effector cells? Can it be excluded that some of the in vivo effects are mediated by MGCP acting on these cells? Is the recruitment of inflammatory cells (others than Th17 cells) affected by MGCP treatment in any of the models?
2. The authors repeatedly use the phrase “increased IFN- γ production” when referring to the abundance of IFN- γ positive CD4⁺ T cells. This is misleading unless the actual production is measured, which was not the case in any of the experiments described in this study.
3. In some experiments using colitis models, MGCP was applied orally. As the commensal microbiota strongly affects development of colitis, it needs to be investigated if MGCP has a direct effect on microbiome composition to differentiate effects directly on immune cells from indirect effects via the microbiota.
4. In the CBir model, the authors mention data on lymphocyte infiltration into the colon and refer to Fig. 2, but there is no data for this presented in the figure. Furthermore, the difference in epithelial hyperplasia is not evident from the micrographs and should be quantified.
5. Line 263: No cytokines were measured, thus the conclusion that MGCP “suppresses pathogenic cytokine production” is not sufficiently supported by data.
6. The antibody blocking experiments (Fig. 6b) lack a control that confirms successful blocking.
7. The authors conclude that while Dectin-1 on DCs is important for Treg induction (based on in vivo results), the suppression of Th1 cells is mediated by TLR2 by a different mechanism (based on in vitro experiments). An alternative explanation for these results regarding dectin-1 deficiency is that not DCs, but another type of immune cells is important for Treg induction in vivo. This should be addressed.

Minor points:

1. There are several minor grammar and other language issues that should be corrected. I recommend proof reading by a native speaker. Also, I strongly suggest to avoid using the term “dramatic” to describe differences – indicate the X-fold change if you want to highlight the magnitude of the difference.
2. Please indicate the source of the antibodies used.
3. Why is the data presented as mean \pm SEM rather than mean \pm SD?

A point-by-point response to reviewers' comments:

In the beginning we thank the reviewers for their time to carefully review our findings and provide us the opportunity to submit a revised version of our manuscript NCOMMS-20-22599-T entitled "Commensal yeast cell surface polysaccharide activities unleashed upon removal of beta-1,3-glucans restrain inflammatory disorders". We have taken into account all the reviewers' comments and performed further experiments to address them. Following reviewers' suggestions, we have also clarified and discussed some of the data presented in the manuscript, and we have modified the text where necessary. Please, find below a detailed point-by point reply to the reviewers.

Reviewer 1

General comment:

After analyzing the manuscript entitled "Commensal yeast cell surface polysaccharide activities unleashed upon removal of β -1,3-glucans restrain inflammatory disorders", which aimed to develop immunomodulatory activities of these polysaccharides after removal of β -glucan- (1,3) - linked by enzymatic action. It is an excellent research and can be published, but the part related to chemical structures must be revised, while the part of biological activity is well developed and with innovative results and must add to those of other research. Some suggestions and questions are presented below

Response to general comment:

We thank the reviewer for carefully reviewing our manuscript, and giving us insightful suggestions for improving it. We have now performed new experiments related to chemical structures related to cell surface polysaccharides, and also modified the text at places as recommended by the reviewer.

Specific comment 1:

The Abstract is presented in a generally extremally discursive and little orientative way, therefore bringing some experimental results obtained, mainly the significant and comparative ones with respective percentages in several experiments that will bring greater orientation to the readers

Response to specific comment 1:

We agree with the reviewer that we presented the abstract with a non-technical undertone. Following the reviewer's suggestion, we have made changes in the abstract as much permissible within the relatively short ~150 words limit, keeping in mind to highlight the broader implications of the findings for the general readership of the journal. We would like to humbly point out that in the "instruction for authors" the journal recommends an abstract with "...a general introduction to the topic and a brief non-technical summary of your main results and their implications".

Specific comment 2:

Between lines 86 to 92 of the "Introduction section" bring precise information on the chemical structures of the polysaccharides present in the cell wall of this yeast and better report the biological action of zymosan in relation to the immune system

Response to specific comment 2:

We appreciate the reviewer's suggestion. We have now expanded the section in the introduction that was pointed out by the reviewer and provided detailed information regarding zymosan structure and their functionality (highlighted text, lines 88-96 of the revised manuscript).

Specific comment 3:

The extraction and purification process (materials and methods) was developed in this research, so the extraction process must be presented in the text, the presentation is incomplete and must be reorganized

Response to specific comment 3:

We thank to reviewer for this suggestion. In order to make the purification process clear to the readers, we added a brief schematic of the polysaccharide purification protocol in Fig. S2a and described it in detail in the Methods section.

Specific comment 4:

Add some lines in the methodology (analysis of the composition of Polysaccharides) developed to obtain the composition of monosaccharides. The way it is presented brings many doubts to the understanding, mainly the process of methylglycosides and patterns used.

Response to specific comment 4:

As suggested by the reviewer, we have provided more details in the “Polysaccharide composition analysis” section (highlighted text, lines 523-533 of the revised manuscript). Please note that this section also includes the method used to analyze the linkage pattern of the residues. This analysis was performed in response to query 6 of Reviewer #2.

Specific comment 5:

Present 1-NMR or 13C-NMR spectra of the polysaccharides present in (a) purified yeast extract; (b) polysaccharide after removal of β -glucan by zymolase; (c) residual polysaccharide + zymosan. In this way, we will have a general view of the chemical structures obtained from the yeast extract. Thus, by integrating the anomeric peaks (1H-NMR) of these structures, we will have an orientation of the type of units of the remaining polysaccharides. 13 C-NMR observe the anomeric signals and the linkage region.

Response to specific comment 5:

This is indeed a very appropriating suggestion to draw comprehensive picture of the chemical structures obtained from yeast extract. We have proton NMR and other spectra data (COSY, TOCSY, NOESY, HSQC, HMBC, HSQCTOCSY) of purified yeast extract, MGCP and have included this in the revised manuscript in Supplementary Figure 2b and c and Supplementary Table 1. The corresponding text is highlighted in lines 161-166. We did not measure the ¹³C spectrum because the NMR instrument has a reverse probe, thus it does not perform well with this sequence. On the other hand, the full set of spectra recorded gives a clear picture of the mixture isolated. The HSQC, HSQCTOCSY (Fig. S2b) and HMBC (Fig. S2c) are reported hereafter with the full labeling. Please note that the residues of the mannan (A, B, C...) are drawn without respecting the proportion that exist among them. In other words, they are not in a 1:1 ratio. The figure of the mannan is given just to explain in which kind of linkage these units are involved.

Unfortunately, we did not measure the spectra of (b) and (c) samples. The reason for this is because the NMR analysis on the yeast extract did not detect β -(1,3)-glucan, nor was it detected by linkage analysis (please also see the response to query 6 of Reviewer #2). We suppose that this component went lost during the purification or, if any left, was below the NMR detection limit. For this reason, at the time of the experiments we assumed that the amount of this component was very low-if not completely absent- and although we performed the β -(1,3)-glucanase treatment, we did not check the proton NMR profile to detect any change. We apologize for not being able to provide this information now; the sample has been used up for the bioassays, and

there is nothing left to measure a proton NMR. We are not able to prepare it once more, since due to the viral pandemic, the access to the laboratories is limited in the countries of all authors. As for the zymosan sample after the treatment with β -(1,3)-glucanase, we did not measure the proton spectrum at the time of the experiment because we trusted the direction given from the manufacturer. We sincerely hope that the Reviewer will accept our apologies in this regard.

Specific comment 6:

I understand that "Figure S1. Chemical and functional properties of MGCP" must be present in the text of the manuscript and not as supplementary material. Here are some important results for understanding the changes related to the chemical structures presented here.

Response to specific comment 6:

We thank the reviewer for this suggestion. We believe that the former Fig. S1d represented a key data in order to dissect the functional properties of the structural components of MGCP. We have now moved this figure in the main Fig. 2f (highlighted text, lines 187-188 of the revised manuscript).

Specific comment 7:

The results related to the different chemical structures should be better discussed and clarified in "Results" and "Discussion".

Response to specific comment 7:

As mentioned above and also in the following points, we have added new data describing chemical structures and discussed and clarified them in the "Results" and "Discussion".

Specific comment 8:

Observing the text of lines 387 to 392 regarding the polysaccharide MGCP is confused because when viewing Figure S1 (E) it shows a B-Glucana-(1,6)-linked with single ramification - (1,3)-linked, however the Figure S1 (a) and (c) show abundant values of mannose. Thus, it brings doubts and the text must be revised

Response to specific comment 8:

We thank the Reviewer for this suggestion and apologize for not making it clear in the discussion. As noted from the Reviewer, mannan is the abundant component in MGCP, while β -1,6-glucan is

15% by weight as calculated by monosaccharide composition. However, even though the glucan is a minor component, it is essential for the functionality of MGCP mixture (Fig 2f-h and Fig S2d in revised manuscript), because our acquired data suggest that this component is the one responsible for the anti-inflammatory activities. Regrettably, we have not been able to isolate the glucan in a pure form. As suggested by the reviewer, we have clarified this issue in the Discussion (highlighted text, lines 454-458 of the revised manuscript).

Specific comment 9:

The extraction and purification process should be improved and an extraction and purification scheme can be added in Figure S1. This scheme should clarify to readers the modifications developed in yeast polysaccharides

Response to specific comment 9:

We thank the reviewer for this suggestion. As mentioned in Comment# 3, we have added a purification scheme in the new Fig. S2a.

Specific comment 10:

In Figure 2 yeast cell wall derived MGCP, in items (b) and (c) the information regarding the structures of MGCP are not correct, because they are not structures produced by NMR, these are structures obtained from literature. In this manuscript the necessary NMR spectra were not presented and there are no experiments to confirm these chemical structures.

Response to specific comment 10:

We appreciate this comment and apologize for not including the NMR data in the original version of the manuscript. Although we relied on published data, we have checked the identity of the polysaccharides in MGCP through NMR analysis, in order to be sure that MGCP contained the polysaccharides that were described. We avoided the inclusion of the NMR spectra in the original version of manuscript because the structures of these components are well known.

In response to the reviewer's comment we have now included the NMR data (Fig. S2b, c) along with the corresponding table of chemical shifts (Table S1). 2D NMR analysis enabled us to evaluate the glucan composition of MGCP, which is reported in the structures of Fig. 2. For the glucan component, the integration between the anomeric densities in the HSQC spectrum (L1 versus N1) was 18 and 82%, respectively, and it indicated the branching degree of the glucan. A similar approach was used for the mannan component, by combining the information from the proton and the HSQC spectra. Please note that for the 2D spectra only, temperature was raised

to 310K to observe one of the glucose signals (label L in the 2D spectra) that otherwise would have been hidden from the residual water signal. Hence, the structures of the two polysaccharides in MGCP are examined by NMR basis, using traditional and established approach. Accordingly, we have added the conditions used to measure the NMR spectra and a brief description of the strategy used to analyze them in the Methods section (highlighted text, lines 553-573 in the revised manuscript).

Specific comment 11:

Bring a better discussion in "Results" and also in "Discussion" section related to Figure 1. Zymosan β -1,3-glucan. In an attempt to clarify the results observed with Zymosan and zymosan together with β -(1,3)-glucanase.

Response to specific comment 11:

As suggested by the reviewer, we have now rearranged, edited and added new text clarifying the results obtained with zymosan and β -(1,3)-glucanase treated zymosan, and discussed the implications of these findings (highlighted text, lines 137-141 and lines 425-431 of the revised manuscript).

Reviewer 2

General comment:

The authors are addressing an interesting aspect of immune-modulatory properties of components of the yeast cell wall. They succeed in some respects especially the biochemistry, but some controls are missing and conclusions for the Treg/Th1 component is not strong and the Cox2 as a mechanism is insufficient. The mouse models show an effect but possibly not for the reasons they state.

Response to general comment:

We thank the reviewer for carefully reviewing our manuscript, and providing insightful comments, addressing which have helped us to substantially improve our manuscript. We have performed new experiments to address the reviewer's comments and suggestions. Our point-by-point response to the reviewer's comments are listed below.

Specific comment 1:

Fig1. The authors should state in text the inclusion of GM-CSF (not clear why they would need this for polarisation of iTregs or Th1). The conclusion that B 1,3 Glucan enhances Th1 and restrains iTregs is a bit misleading. In fact, one might argue that there's no effect on iTreg induction with their no treatment group, Th0 control is required, as their inducing conditions actually shows no Foxp3. 1b, Needs control. Th0, and preferably Th17 as the interplay with Tregs is at the center of this story. The effect of Th1 induction by B 1,3 Glucan at 200ug is modest but believable. However, in Fig 1d, to claim suppression here is inaccurate. The authors should show expression of Foxp3 in this figure (i.e is decreased percentage due to induction of Foxp3 or actual antagonism of Tbet programme). Controls of Th0 and Th17 would also support their claim.

Response to specific comment 1:

We thank the reviewer for pointing this out and apologize for not making this clear before. We added low amount of GM-CSF in our *in vitro* experiments, in anticipation that it would promote DC viability during culture. We have mentioned this in the Methods section of the revised version of the manuscript (highlighted text, line 627 of the revised manuscript). Furthermore, we have provided our results upon side by side comparison, which indicates that there is no difference in Treg or Th1 generation with or without GM-CSF under respective differentiation conditions (Rebuttal Fig.1 for reviewer).

We have performed Th0 and th17 controls and presented the data in the new Fig. S1. Interestingly for the FACS plots where IFN- γ was analyzed, the base line of IFN- γ expression in the cells skewed under Th1 condition was consistently shifted towards right, when compared with Th0 conditions, making it difficult to identically gate the IFN- γ expressing populations. Nevertheless, the results indicate that specifically under Th1 skewing condition, zymosan treatment leads to moderate but significant increase Th1 differentiation, as determined by the increase in frequencies of CD4⁺IFN- γ ⁺ T cells (New Fig. S1a and Fig. 1a). We have described these results in the highlighted text, lines 116-125 of the revised manuscript.

As suggested by the reviewer, we have also analyzed the expression of Foxp3 in Fig. 1c, which was found to remain largely unaffected under the conditions employed.

Specific comment 2:

Fig2. Shows nicely the IFN issue discussed above (see Fig2e. The x-axis, cell size moves closer to zero).

Response to specific comment 2:

As mentioned in the response to specific comment 1, we consistently observed that the baseline of IFN- γ expressing populations to be shifted towards the right under Th1 skewing conditions. Upon MGCP treatment under identical Th1 conditions, the frequency of IFN- γ^+ population as well as the baseline of IFN- γ expression was reduced in a dose dependent manner, resulting in the shift of the negative population towards the left.

Specific comment 3:

Fig3. Is not convincing. In general this comment goes to other autoimmune models in this paper. MGCP induction of de novo iTregs may be happening. However, because Tregs regulated antigen specific responses at a DC level, the reduced Th1 response observed is likely due to Lack of antigen (Ethan Shevak's Lab showed this, Nat Immunology 2019, Akkaya).

Response to specific comment 3:

In the study mentioned by the reviewer, the authors have very elegantly shown that one mechanism of repression by antigen specific Treg cells is to directly bind dendritic cells and inhibit T naïve cells, specific for the cognate antigen, from getting access to peptide MHCII complex on dendritic cells. The authors have also shown that in this process, the strong interaction between antigen specific Treg cells and DCs leads to reduction of MHCII from DC surface, resulting in compromised antigen presentation by DCs.

In light of this study, we assessed the expression of MHCII on DC surface in draining lymph nodes of mock treated or MGCP treated mice after induction of EAE. We observed that MHCII expression remained undiminished, even slightly increased on DC surface in the draining lymph nodes of MGCP treated mice. It therefore appears unlikely that the reduction in Th1 response observed in MGCP treated groups in the autoimmune models employed is solely a consequence of lack of antigen presentation to the naive T cell population. We have discussed this observation in the new Fig. S6i (highlighted text, lines 287-296 of the revised manuscript).

Specific comment 4:

Fig4. This is consistent with their Data in Figure 4 (EAE model) which shows that there's no accumulation of iTregs in the CNS but rather in dLN and Fig4 I shows nicely the BOTH antigen-specific TH1 and Th17 and also the dual producers are both downregulated, making the claim about regulation of Th1 weak.

Response to specific comment 4:

As indicated in the response to specific comment 3, we believe that lack of antigen presentation in draining lymph nodes, although may be one of several mechanisms of Treg mediated suppression, is not likely the only reason for reduced Th1 differentiation in MGCP treated mice under conditions of EAE. We agree with the reviewer that the mechanism of MGCP mediated Th1 as well as Th17 populations in the context of EAE is not completely clear, and multiple mechanisms of suppression may be at play. We have humbly acknowledged this issue and added a section addressing this point in the discussion (highlighted text, lines 435-441 in the revised manuscript).

Specific comment 5:

The authors should demonstrate that MGCP does not work during an inflammatory model in dectin-1 mice (eg: EAE)

Response to specific comment 5:

As suggested by the reviewer, we have performed EAE experiments in Dectin1 sufficient and deficient mice that were either mock treated or treated with MGCP. Indeed, we observed that the beneficial effects of MGCP in the experimental EAE model is completely abolished in Dectin1 deficient mice. We have presented this data in the new Fig. 7e, f and corresponding highlighted text, lines 379-386 in the revised manuscript.

Specific comment 6:

The authors should show that MGCP is not present in depleted zymosan, which lacks TLR2 stimulatory activity (PMID: 12719479)

Response to specific comment 6:

We really appreciate the reviewer's insightful suggestion. We have analyzed zymosan and depleted zymosan by proton NMR analysis and found that the β -glucan is present in depleted zymosan, while mannan is absent. However, we noted that both zymosan and depleted zymosan gave a precipitate at the concentration used for recording proton NMR spectra, indicating that the NMR spectra of both polysaccharides report only a soluble component. However, unlike zymosan and depleted zymosan, MGCP is completely soluble so that the NMR results can be considered representative of the polysaccharide content.

In order to obtain a complete picture of MGCP, zymosan and depleted zymosan, we performed linkage analysis to distinguish 3-linked glucose (representative of β -1,3-glucan) from 6-linked glucose (representative of β -1,6-glucan), and quantified these two different linkages by comparing the areas of their representative derivatives. We found that both normal and depleted zymosan contain 3- and 6-linked glucose, and also that there was a net increase of 6-linked glucose (or β -1,6-glucan) in depleted zymosan.

Lastly, the total amount of the glucans including both β -1,3- and β -1,6-glucan is rather high in zymosan, while mannan is totally absent (or below the detection limit) in the depleted zymosan. In the chromatogram of zymosan we could find only terminal mannose. Terminal sugars generally respond better with GC-MS detector in comparison to their substituted forms, and for this reason, among units that compose mannan, we could detect only the terminal mannose, and not the others. Nevertheless, the presence of mannan has been prominently revealed by proton NMR analysis of the soluble fraction of the zymosan, in agreement with what is expected based on the nature of the sample. Interestingly, MGCP analysis result revealed presence of mannan and specifically β -1,6-glucan. β -1,3-Glucan was completely undetectable. We have presented the data in a new supplementary figure (Fig. S4) and described the findings in highlighted text, lines 206-218 of the revised manuscript.

Reviewer 3

General comment:

The manuscript by Rudra et al. describes a soluble yeast cell wall preparation (MGCP) with immunomodulatory properties. Using in vitro and several in vivo models, the authors convincingly demonstrate the capacity of MGCP to induce Tregs by modulating DC function, and to suppress the generation of IFN- γ producing CD4⁺ cells. This correlated with significantly reduced disease severity in several models of inflammatory diseases. They furthermore showed that Dectin-1 and TLR2 are involved in the effect of MGCP on DCs and T cells.

These findings are novel and very interesting both from a biological perspective and due to the possible implications for novel therapeutic approaches.

The experiments are technically sound. There are however some points that need to be addressed:

Response to general comment:

We thank the reviewer for reviewing our manuscript and suggesting insightful propositions for its improvement. Our point by point response to the reviewer's comments are listed below.

Specific comment 1:

The authors focused their in vivo analyses on CD4+ T cells– what about APCs, and myeloid effector cells? Can it be excluded that some of the in vivo effects are mediated by MGCP acting on these cells? Is the recruitment of inflammatory cells (others than Th17 cells) affected by MGCP treatment in any of the models?

Response to specific comment 1:

We thank the reviewer for this suggestion. In order to address the reviewer's comment, we have analyzed the frequencies of Macrophages, monocytes and Monocyte-derived DCs (MoDCs) in the spinal cord of mock treated or MGCP treated mice after induction of EAE. We observed, while the monocyte population varied significantly upon EAE induction and was rectified, that of Macrophage and MoDCs remained largely unchanged in MGCP treated group. The results are described in the new Fig. S6h (highlighted text, lines 284-286 in the revised manuscript).

Specific comment 2:

The authors repeatedly use the phrase “increased IFN- γ production” when referring to the abundance of IFN- γ positive CD4+ T cells. This is misleading unless the actual production is measured, which was not the case in any of the experiments described in this study.

Response to specific comment 2:

As suggested by the reviewer, we have replaced the phrase “IFN- γ production” with “Th1 differentiation” throughout the manuscript.

Specific comment 3:

In some experiments using colitis models, MGCP was applied orally. As the commensal microbiota strongly affects development of colitis, it needs to be investigated if MGCP has a direct effect on microbiome composition to differentiate effects directly on immune cells from indirect effects via the microbiota.

Response to specific comment 3:

We appreciate the reviewer for suggesting this experiment, since it is well described that microbiome affects manifestation of multiple inflammatory diseases including colitis. We induced colitis through adoptively transferring naïve CD4 T cells isolated from congenically marked (CD45.1) Foxp3^{EGFP} mice and orally administered mock (DW) or MGCP every other day starting from the day of cell transfer. MGCP supplementation was continued until the end of experiment. To analyze microbiome landscape, we collected stool on the day before transferring cells (D0) and endpoint of the experiment (D32). As expected, there were significant differences between microbial composition between D0 and D32 samples in both mock and MGCP treated groups. However, there was no significant MGCP dependent change in microbiome composition in D32 samples compared to mock. This result suggests that MGCP treatment has little effect on microbial composition under inflammatory condition. These results are presented in Fig. S8a-c and discussed in highlighted text, lines 344-357.

Specific comment 4:

In the CBir model, the authors mention data on lymphocyte infiltration into the colon and refer to Fig. 2, but there is no data for this presented in the figure. Furthermore, the difference in epithelial hyperplasia is not evident from the micrographs and should be quantified.

Response to specific comment 4:

As suggested by the reviewer, we have recalculated histopathological score more carefully. Also, we included new representative histology sections comparing pathologic symptoms of mock treated versus MGCP treated colon, where the lymphocytic infiltrations in the “mock” are visible more prominently. For the reviewer’s convenience we have also included larger versions of the histology pictures as Rebuttal Fig. 2. These results are presented in Fig. 5d-e in the revised version of manuscript. We have also added a section in the Methods section, describing the criteria followed to calculate the histology scores (highlighted text, lines 743-750 of the revised manuscript).

Specific comment 5:

Line 263: No cytokines were measured, thus the conclusion that MGCP “suppresses pathogenic cytokine production” is not sufficiently supported by data.

Response to specific comment 5:

We thank the reviewer for this comment. We have replaced the mentioned phrase by “... suppressing the generation of pathologic cytokine producing immune cell populations.” (highlighted text, line 298 in the revised manuscript).

Specific comment 6:

The antibody blocking experiments (Fig. 6b) lack a control that confirms successful blocking.

Response to specific comment 6:

We used concentration of blocking antibody in the experiments according to technical data sheet provided from manufacturer. According to manufacturer's instruction, a concentration of 1µg/ml to 10µg/ml of anti-Mannose receptor and anti-Mincle antibody is sufficient to block the corresponding cognate receptors, while 5µg/ml is sufficient for blocking of DC-SIGN. In the original version of manuscript, the maximum antibody concentration of anti-Mincle and anti-Mannose receptor we used was 1µg/ml and it was minimum amount based on technical data sheet. In order to confirm our results, we have now performed same experiment with the maximum recommended dose of 10µg/ml. These revised results are presented in Fig. S9a (line 372) in current version of manuscript.

Specific comment 7:

The authors conclude that while Dectin-1 on DCs is important for Treg induction (based on in vivo results), the suppression of Th1 cells is mediated by TLR2 by a different mechanism (based on in vitro experiments). An alternative explanation for these results regarding dectin-1 deficiency is that not DCs, but another type of immune cells is important for Treg induction in vivo. This should be addressed.

Response to specific comment 7:

We thank the reviewer for this comment. We would like to point out that in addition to the in vivo experiments described in Fig. 6d (Fig. 7d in the revised manuscript), we also directly addressed the role of Dectin1 on DCs with regard to their MGCP dependent Treg inducing capacity under in vitro culture conditions. We found that the MGCP dependent Treg inducing capacity of DCs is severely compromised in Dectin1 deficient DCs (Fig. 7a). In order to determine whether other antigen presenting cells are also capable of promoting MGCP dependent Treg induction, we further tested whether macrophages can promote Treg induction in the presence of MGCP. In

this experiment presented in revised Fig. S7a (described in highlighted text, lines 309-314), we found that unlike DCs, macrophages have superior Treg inducing capacity at steady state, however this already enhanced activity is only marginally increased upon MGCP treatment. These results, in conjunction with Figures 6c-e and 6i strongly indicate that DCs are the primary cell populations responsible for MGCP mediated enhanced induction of Treg cells.

Minor comment 1:

There are several minor grammar and other language issues that should be corrected. I recommend proof reading by a native speaker. Also, I strongly suggest to avoid using the term “dramatic” to describe differences – Indicate the X-fold change if you want to highlight the magnitude of the difference.

Response to minor comment 1:

We thank the reviewer for the suggestion and pointing this out. In the revised manuscript, proof reading was also performed by a native speaker. Also we have removed the term “dramatic” wherever applicable.

Minor comment 2:

Please indicate the source of the antibodies used.

Response to minor comment 2:

We apologize for not providing detailed antibody information in the original manuscript. We have now listed the source of each antibody in a new Supplementary Table 2 of the revised manuscript (mentioned in highlighted text 604-606).

Minor comment 3:

Why is the data presented as mean \pm SEM rather than mean \pm SD?

Response to minor comment 3:

We agree with the review's suggestion that variation of data among individual experiments or animals are better represented by \pm SD. For most of the data, except for colitis weight loss and EAE scores, we have now changed the presentation to \pm SD.

Rebuttal Figures:

Rebuttal Fig. 1

(a-b) Splenic CD11c⁺ DCs were stimulated with indicated polysaccharides and cultured with naïve CD4 T cells in the presence or absence of GM-CSF along with minimal amount of Th1 (a) or Treg (b) driving cytokines. Representative FACS plots (left) and frequencies (right) of IFN- γ ⁺ CD4 T cells (a) and CD4⁺Foxp3⁺ T cells (b) are shown. All graphs show the mean \pm SD. *p < 0.05, **p < 0.01, ***p < 0.001, ****p < 0.0001 (Student's t test).

Rebuttal Fig. 2

Larger versions of representative histology sections that are shown in Fig. 5d, where lymphocyte infiltration in mock treated sample is more prominently visible.

REVIEWER COMMENTS

Reviewer #2 (Remarks to the Author):

I am happy with the revised changes to the manuscript, and the new data have addressed my concerns. It would be good if the authors could speculate how Dectin-1 signalling could have such disparate effects on DC function.

Additional response to reviewer 1 previous concerns:

To me, reviewer one was asking for more detail on components and was not really criticising the results. In my opinion, the authors have satisfactorily addressed all of the concerns of reviewer 1, and included all the requested information (barring 2 small samples, which I thought they explained well) in the revised manuscript.

Reviewer #3 (Remarks to the Author):

The authors have sufficiently addressed the points raised by performing additional experiments and revising the manuscript.

REVIEWERS' COMMENTS

We thank the reviewers for their insightful suggestions before that helped us develop our manuscript, and we are happy that we were able to address the reviewers' concerns.

Reviewer #2 (Remarks to the Author):

I am happy with the revised changes to the manuscript, and the new data have addressed my concerns. It would be good if the authors could speculate how Dectin-1 signalling could have such disparate effects on DC function.

Authors' response:

We have included our thoughts on the disparate effect of Dectin1 on DC function in the current version of the manuscript in line 471-477 highlighted with blue colour.

Additional response to reviewer 1 previous concerns:

To me, reviewer one was asking for more detail on components and was not really criticising the results. In my opinion, the authors have satisfactorily addressed all of the concerns of reviewer 1, and included all the requested information (barring 2 small samples, which I though they explained well) in the revised manuscript.

Authors' response:

We thank the reviewer for the additional response on reviewer 1's concerns and happy that we were able to satisfactorily able to address those points.

Reviewer #3 (Remarks to the Author):

The authors have sufficiently adressed the points raised by performing additional experiments and revising the manuscript.

Authors' response:

We are happy to hear that reviewer 3 is satisfied with our revised manuscript.